# Global Sensitivity in Private Policy Selection with Nuisance-Fitted Doubly Robust Scores

## Abstract

Doubly robust (DR) scores are commonly used to compare policies from observational data. In a private release, one may fit nuisance models inside a trusted environment and apply the exponential mechanism only to the final policy choice. The usual constant-sensitivity calculation for a sum of bounded scores is then incomplete: replacing one nuisance-training record can change the fitted score at every record in a disjoint scoring sample. We formalize this training-to-scoring contribution for selection from a finite public policy library. It can be proportional to the scoring-sample size even when the propensity model is exact, the outcome-regression error vanishes, and the population DR value is unbiased. Thus, standard DR accuracy guarantees do not determine the global sensitivity of the learned utility. We give a sufficient pure-DP calibration from a public uniform replace-one prediction-stability bound and derive a regret bound separating statistical error from privacy cost. The bound is learner-specific and can be conservative. Our experiments report sampled sensitivity audits for flexible learners and an empirical risk minimization (ERM) positive control calibrated from public deterministic stability constants.

## 1 Introduction

Data-driven policies increasingly guide treatment decisions, public-service allocation, and online interventions (Murphy, 2003; Qian & Murphy, 2011; Zhao et al., 2012; Kitagawa & Tetenov, 2018; Athey & Wager, 2021). In observational studies, their values must be estimated despite nonrandom treatment assignment and missing counterfactual outcomes. Doubly robust (DR) scores are widely used for this purpose. They combine an outcome regression with a propensity model, and their population error is governed by a second-order product remainder under the usual orthogonality conditions (Dudík et al., 2011; Kallus, 2018; Zhao et al., 2019; Chernozhukov et al., 2018). This structure allows policy evaluation to use flexible nuisance models while reducing the first-order effect of nuisance estimation error on policy-value comparisons.

Privacy changes how this pipeline can be deployed. Medical records, administrative data, and platform logs may be available inside a trusted data custodian, while the resulting estimate or decision must be released outside it. Existing work has already combined DP with DR and orthogonal causal learning. Niu et al. (2022), for example, build private CATE meta-learners, including DR and R learners, from differentially private base learners and sample splitting. DP-CATE (Schröder et al., 2025b) similarly places the relevant stages of an orthogonal CATE learner and its released effect function inside the privacy analysis. These approaches provide end-to-end guarantees, but require DP-compatible training procedures and may restrict the available nuisance learners or introduce additional estimation error before the downstream causal object is formed.

A complementary approach is to retain non-private nuisance fitting inside the trusted environment and privatize only the downstream release. This preserves access to flexible outcome and propensity models and, through the DR construction, their statistical benefits for policy-value comparison. Lebeda et al. (2025) follow this route by protecting nuisance predictions and fold-level aggregation for scalar causal estimands. PrivATE (Schröder et al., 2025a) releases DR estimates, variances, and confidence intervals through

influence-function and smooth-sensitivity arguments. These results make the same design attractive for policy selection: when deployment is restricted to a prespecified menu of treatment protocols, eligibility rules, or operational policies, the analyst may need to release only the identity of the selected policy.

However, existing privacy analyses do not directly cover this release architecture. Downstream causal-estimation methods protect scalar estimates or effect functions, while private treatment-rule methods privatize the optimization and release of a learned rule. Direct selection from a public policy library instead applies the exponential mechanism to comparison utilities constructed from nuisance-fitted DR scores. The same nuisance fitting that improves statistical policy evaluation also creates an additional privacy dependence: replacing one nuisance-training record may change the fitted score map at every scoring covariate and hence alter all utility summands. Bounded scores control the movement of each summand, but not the number of summands that move. Sample splitting prevents direct record reuse, and standard DR guarantees control population bias and nuisance-error remainders, but neither provides the finite-sample global-sensitivity bound required by the exponential mechanism. This leads to the central question:

> *How does nuisance fitting affect the global sensitivity of DR policy utilities, and what conditions suffice for valid private policy selection?*

We answer this question by isolating the sensitivity introduced by nuisance fitting. We show that favorable DR population guarantees can coexist with utility sensitivity linear in the scoring-sample size, and then give a sufficient pure-$\varepsilon$-DP calibration based on a public uniform replace-one prediction-stability bound. The resulting regret analysis separates statistical error from the additional privacy cost induced by the learned score map. The experiments examine the negative mechanism through sampled sensitivity audits for flexible boosted-tree learners and instantiate the positive result using an empirical risk minimization (ERM) learner with public deterministic stability constants.

Our contributions are as follows:

- **A release-specific sensitivity decomposition.** We distinguish fixed, frozen, and learned score maps and decompose the global sensitivity of the learned DR utility into a direct scoring-record term and a nuisance-fitting term transmitted across the scoring block.

- **A DR-specific separation.** We prove that exact propensity estimation, vanishing outcome-regression error, zero population DR bias, bounded realized scores, and orthogonal product-remainder control can coexist with utility sensitivity proportional to the scoring-sample size.

- **A stability-based calibration and regret guarantee.** A public uniform replace-one prediction-stability bound yields a sufficient pure-DP calibration and a regret bound separating statistical and privacy terms. The experiments use sampled adjacent-pair audits for boosted trees and an ERM positive control calibrated from public deterministic stability constants.

## 2 Related Work

**DR policy evaluation and learning.** Outcome regression, inverse propensity weighting, and DR scores are standard tools for evaluating and optimizing policies from observational data (Qian & Murphy, 2011; Zhao et al., 2012; Kitagawa & Tetenov, 2018; Kallus, 2017; 2018; Zhao et al., 2019; Athey & Wager, 2021). Orthogonality reduces the leading population effect of nuisance-estimation error, typically to a product remainder (Chernozhukov et al., 2018). These results justify flexible nuisance modeling for statistical accuracy. They do not provide a worst-case replace-one bound for the realized score map.

**Private DR and orthogonal learners.** DP and DR estimation have already been combined. Niu et al. (2022) give a private CATE meta-algorithm covering DR and R learners through sample splitting, parallel composition, and differentially private base learners. DP-CATE (Schröder et al., 2025b) develops a Neyman-orthogonal framework whose formal privacy analysis includes DP-compatible nuisance estimation and the released CATE queries or function. Such stage-wise approaches provide complete accounting. In practice,

they require a DP-compatible optimization procedure, may constrain the usable learner class, and incur privacy or optimization costs before the final release.

**Non-private nuisances with downstream privatization.** Other work separates nuisance fitting from the privacy mechanism. Lebeda et al. (2025) fit arbitrary nuisance models non-privately and protect their predictions and fold-level aggregation when releasing G-formula, IPW, or DR estimates. PrivATE (Schröder et al., 2025a) releases DR estimates, variances, and confidence intervals through influence-function arguments and uses gross-error sensitivity to upper-bound smooth sensitivity. Its control of the first-stage nuisance contribution assumes bounded nuisance predictions, standard product-rate conditions, and a local sup-norm neighborhood of the true nuisance functions. Our theorem addresses a different release object and sensitivity target: it requires a public finite-sample uniform replace-one bound on nuisance predictions to calibrate the global sensitivity of the policy-comparison utility used by the exponential mechanism.

**Private treatment-rule and decision learning.** Several methods directly privatize treatment-rule learning. PrOWL (Spicker et al., 2024) constructs a private outcome-weighted learner for dynamic treatment regimes; Giddens et al. (2026) develop differentially private weighted ERM and instantiate it for outcome-weighted learning; and DP-2ERM (Lee & Chen, 2026) calibrates second-stage private ERM to data-dependent perturbations of first-stage balancing weights. These methods release a learned rule or model parameter through private optimization. Our setting instead selects one policy from a fixed public library with an exponential mechanism applied to nuisance-fitted DR utilities, so the relevant object is the global sensitivity of the comparison utility rather than the parameter sensitivity of an optimized rule.

**Private selection and stability.** The exponential mechanism relies on a global-sensitivity bound, while private ERM privatizes the optimization procedure itself (McSherry & Talwar, 2007; Dwork & Roth, 2014; Chaudhuri et al., 2011; Bassily et al., 2014). More broadly, replace-one behavior of data-dependent learning procedures is a standard concern in differential privacy and algorithmic stability (Bousquet & Elisseeff, 2002). Against that background, our contribution is specific to learned DR policy utilities: the nuisance-fitting sensitivity decomposition, the DR-specific separation, and a sufficient calibration for direct selection. Private contextual-bandit and reinforcement-learning methods protect the learner, sufficient statistics, or interaction history (Shariff & Sheffet, 2018; Garcelon et al., 2022; Qiao & Wang, 2023a;b). Smooth or local sensitivity provides another route under approximate DP when a valid data-dependent bound is available (Nissim et al., 2007).

## 3 Problem Setup

**Policy value and private policy selection.** We use the potential outcomes framework. Consider a dataset of $N = m + n$ records partitioned into a fixed, public, and data-independent split: $D = (D^{\mathrm{tr}}, D^{\mathrm{sc}})$, $D^{\mathrm{tr}} = \{Z_1, \ldots, Z_m\}$, and $D^{\mathrm{sc}} = \{Z_{m+1}, \ldots, Z_{m+n}\}$. We assume each record $Z_i = (X_i, A_i, Y_i)$ is drawn independently from a population distribution $P$. Throughout, $Pf$ denotes $\mathbb{E}_P[f]$. Here, $X_i \in \mathcal{X}$ denotes the covariate vector, $A_i \in \mathcal{A} = \{1, \ldots, K\}$ the discrete action, and $Y_i \in [\underline{y}, \overline{y}]$ the bounded observed outcome. Let $Y(a)$ denote the potential outcome under action $a$. We make the standard causal assumptions of consistency, $Y = Y(A)$, and unconfoundedness, $Y(a) \perp A \mid X$ for all $a \in \mathcal{A}$. The population outcome regression and propensity score are respectively defined as:

$$\mu_{0,a}(x) = \mathbb{E}[Y \mid A = a, X = x], \qquad e_{0,a}(x) = \mathbb{P}(A = a \mid X = x).$$

To ensure identification, we assume multi-action overlap: there exists a deterministic constant $\zeta \in (0, 1/K]$ such that $e_{0,a}(x) \geq \zeta$ for all $a$ and $x$.

The target is pure-DP selection from a public and data-independent library $\Pi_n = \{\pi_1, \ldots, \pi_{M_n}\}$. The library may be generated using public randomness fixed before the private data are observed. Conditional on that randomness, the realized library is deterministic. Let $\Pi$ be a target policy class and $\Pi_n \subseteq \Pi$ the finite release library, where each $\pi_j : \mathcal{X} \to \mathcal{A}$. The value of a policy is $V_P(\pi) = \mathbb{E}[Y(\pi(X))]$. Let $\pi_P^\star \in \arg\max_{\pi \in \Pi} V_P(\pi)$ denote the population oracle and $\pi_n^\star \in \arg\max_{\pi \in \Pi_n} V_P(\pi)$ the best candidate in the library.

We fit the nuisance functions on protected data and release only the selected policy identity. The nuisance learners are neither released nor trained by a separate DP mechanism, but their data dependence remains

part of the sensitivity of the final release. The positive result therefore requires a public uniform replace-one prediction-stability bound for the fitted nuisance functions.

The public, data-independent library isolates this nuisance-fitting contribution. Our analysis covers private selection from the fixed library. Constructing or adapting the candidates using protected data would be an additional private learning stage.

**The doubly robust score.** We evaluate each policy with the doubly robust (DR) score Chernozhukov et al. (2018). For a given nuisance pair $\eta = (\mu, e)$, the DR score for a record $Z$ under policy $\pi$ is defined as:

$$\psi_\pi(Z; \eta) = \mu_{\pi(X)}(X) + \frac{\mathbf{1}\{A = \pi(X)\}}{e_{\pi(X)}(X)}\{Y - \mu_{\pi(X)}(X)\}. \tag{1}$$

**Assumption 1.** *Throughout the privacy analysis, learned nuisance functions are deterministically projected, and the true nuisance functions lie in the same ranges. Thus, for every training sample $S$, action $a$, and covariate $x$,*

$$\underline{\mu} \le \mu_{0,a}(x), \widehat{\mu}_{S,a}(x) \le \overline{\mu}, \qquad e_0(x), \widehat{e}_S(x) \in \Delta_K^\zeta := \{p \in \Delta_K : \min_{1 \le a \le K} p_a \ge \zeta\}.$$

Under Assumption 1, let $B_\mu = \max\{\left|\underline{\mu}\right|, \left|\overline{\mu}\right|\}$ and $R_{Y,\mu} = \max\{\left|\overline{y} - \underline{\mu}\right|, \left|\underline{y} - \overline{\mu}\right|\}$. Then, for any nuisance pair satisfying the deterministic ranges, $|\psi_\pi(Z; \eta)| \le B_\mu + \zeta^{-1} R_{Y,\mu} =: B_\psi$.

The statistical appeal of DR scores stems from the following identity:

$$P\psi_\pi(\eta) - V_P(\pi) = \sum_{a=1}^K \mathbb{E}\left[\mathbf{1}\{\pi(X) = a\}\left(\frac{e_a(X) - e_{0,a}(X)}{e_a(X)}\right)\{\mu_a(X) - \mu_{0,a}(X)\}\right]. \tag{2}$$

This yields the well-known product remainder bound in causal inference (doubly robust property):

$$|P\psi_\pi(\eta) - V_P(\pi)| \le \zeta^{-1} \sum_{a=1}^K \|\mu_a - \mu_{0,a}\|_{L_2(P_X)} \|e_a - e_{0,a}\|_{L_2(P_X)}. \tag{3}$$

The identity (2) and the product-remainder bound (3) are proved in Lemma 3. We state them here because they are the statistical benchmark against which the privacy-sensitivity results below are contrasted.

**Differential privacy and utility regimes.** We say two split datasets $D = (D^{\mathrm{tr}}, D^{\mathrm{sc}})$ and $D' = (D^{\mathrm{tr}\prime}, D^{\mathrm{sc}\prime})$ are adjacent, denoted $D \sim D'$, if they differ by the replacement of a single record in either the training block or the scoring block. The split is public and index-based: the replaced record remains in the same block under adjacency. A randomized mechanism $\mathcal{M}$ satisfies $\varepsilon$-differential privacy ($\varepsilon$-DP) if, for all adjacent datasets $D \sim D'$ and all measurable events $S$,

$$\mathbb{P}\{\mathcal{M}(D) \in S\} \le e^\varepsilon \mathbb{P}\{\mathcal{M}(D') \in S\}.$$

To privately select a policy from $\Pi_n$, we use the exponential mechanism with utility $U(D, \pi)$. Given a calibration parameter $\Delta$, the mechanism samples

$$\mathbb{P}(\widehat{\pi} = \pi \mid D) \propto \exp\left\{\frac{\varepsilon U(D, \pi)}{2\Delta}\right\}, \qquad \Delta \ge \sup_{D \sim D'} \sup_{\pi \in \Pi_n} |U(D, \pi) - U(D', \pi)|. \tag{4}$$

For causal-inference readers, (4) is a randomized release rule, not a causal model or an estimating equation. Conditional on the observed dataset, each candidate policy receives a weight that increases with its empirical utility $U(D, \pi)$. The normalizing constant over $\Pi_n$ turns these weights into probabilities. The scale $\Delta$ must be a deterministic upper bound on the largest possible change of any policy utility under a one-record replacement. The factor $\varepsilon/(2\Delta)$ is the standard exponential-mechanism calibration: larger utility gaps make high-utility policies more likely, while a larger sensitivity bound flattens the distribution to maintain pure differential privacy. Thus the privacy question reduces to identifying a valid global sensitivity bound for the

realized utility. This is immediate when the score map is fixed before observing the private data, but becomes subtle when the score map itself is learned from protected records. To further distinguish, we consider the following three utility regimes in this paper:

$$U_{\text{fix}}(D^{\text{sc}}, \pi) = \sum_{i \in D^{\text{sc}}} q_\pi(Z_i), \quad U_{\text{fr}}(D^{\text{sc}}, \pi) = \sum_{i \in D^{\text{sc}}} \psi_\pi(Z_i; \eta^\dagger), \quad U_{\text{ls}}(D, \pi) = \sum_{i \in D^{\text{sc}}} \psi_\pi(Z_i; \widehat{\eta}(D^{\text{tr}})).$$

Here $\psi_\pi$ is the record-level DR score, whereas $U_{\text{fix}}$, $U_{\text{fr}}$, and $U_{\text{ls}}$ are policy-level utilities obtained by summing record-level scores over the scoring block. The subscript "ls" denotes a learned score map. Specifically, $U_{\text{fix}}$ is the fixed-score utility, $U_{\text{fr}}$ is the frozen-nuisance DR utility, and $U_{\text{ls}}$ is the learned-score DR utility. The three regimes differ in which parts of the score map are fixed relative to the protected data:

**Regime I: Fixed-score utility** $U_{\text{fix}}$**.** The score map $q_\pi : \mathcal{Z} \to \mathbb{R}$ is bounded and fixed before the private dataset is observed.

**Regime II: Frozen-nuisance DR utility** $U_{\text{fr}}$**.** The nuisance object $\eta^\dagger$ is fixed relative to the protected scoring/release block and is conditioned upon in the privacy analysis. For this regime only, the privacy statement concerns replacements in the scoring/release block alone: the two scoring blocks may differ by one record, while $\eta^\dagger$ is held fixed. Auxiliary records used to construct $\eta^\dagger$ lie outside this DP guarantee unless protected by a separate privacy analysis.

**Regime III: Learned-score DR utility** $U_{\text{ls}}$**.** The nuisance estimator $\widehat{\eta} = (\widehat{\mu}, \widehat{e})$ is learned from the protected training block. The DP guarantee is with respect to the full split dataset $D = (D^{\text{tr}}, D^{\text{sc}})$, so a neighboring change may occur in either block.

## 4    Global Sensitivity of Learned DR Utilities

### 4.1    Sensitivity Decomposition

This section analyzes direct private policy selection. Because nuisance fitting is not separately privatized, the final sensitivity calculation must include its dependence on the protected training block. We first record the fixed-score baseline. If the score map is fixed before the protected data are observed, replacing one scoring record changes one bounded summand. Frozen-nuisance DR scores have the same property after conditioning on the nuisance functions. Appendix A states the corresponding result.

For a learned score map, a replacement in $D^{\text{tr}}$ can alter $\widehat{\eta}(D^{\text{tr}}) = (\widehat{\mu}, \widehat{e})$. The changed nuisance functions are evaluated on every record in $D^{\text{sc}}$, so their prediction movement accumulates across the scoring block. This accumulation is the nuisance-fitting contribution to global sensitivity. For adjacent nuisance-fitting samples $S \sim S'$ of size $m$, define

$$\Gamma_{\mu,m} := \sup_{S \sim S'} \max_a \sup_x |\widehat{\mu}_{S,a}(x) - \widehat{\mu}_{S',a}(x)|, \qquad \Gamma_{e,m} := \sup_{S \sim S'} \max_a \sup_x |\widehat{e}_{S,a}(x) - \widehat{e}_{S',a}(x)|. \tag{5}$$

These quantities measure the largest replace-one change of the fitted prediction functions. They are deterministic stability quantities, distinct from the population estimation errors $\|\widehat{\mu} - \mu_0\|_{L_2(P_X)}$ and $\|\widehat{e} - e_0\|_{L_2(P_X)}$.

**Lemma 1.** *Let $\rho_m := (1 + \zeta^{-1})\Gamma_{\mu,m} + \zeta^{-2} R_{Y,\mu} \Gamma_{e,m}$. Under Assumption 1, the internally learned DR utility has global sensitivity*

$$\Delta_{\text{ls}}^{\text{glob}} := \sup_{D \sim D'} \sup_{\pi \in \Pi_n} |U_{\text{ls}}(D, \pi) - U_{\text{ls}}(D', \pi)| \leq \max\{2B_\psi, n\rho_m\}. \tag{6}$$

The proof of Lemma 1 is stated in Appendix B.

The bound separates the two locations of the replacement. A scoring-block replacement changes one bounded score and contributes $2B_\psi$. A training-block replacement leaves the scoring records fixed but can change each fitted score by $\rho_m$, and summing over $n$ scoring records gives $n\rho_m$.

Bounded DR scores therefore control the contribution of each scoring record, while leaving the data dependence of the score map unresolved. A pure-DP calibration for $U_{\text{ls}}$ needs a deterministic bound on replace-one

nuisance prediction movement. The next result shows that standard DR accuracy conditions do not supply such a bound.

## 4.2 A DR-Specific Separation between Statistical Accuracy and Global Sensitivity

Lemma 1 identifies the missing quantity: a uniform replace-one bound for the learned nuisance predictions. The DR product remainder in (3) controls a population expectation. Differential privacy requires a worst-case bound over realized adjacent datasets.

The following construction separates these quantities. The propensity learner is exact, the outcome-regression error vanishes uniformly over training samples, and the population DR value is unbiased. A one-record replacement can still change the score on a low-probability covariate region. A worst-case scoring block may place all of its records in that region, producing a utility change of size $n$.

**Theorem 1.** *There exist a fixed population distribution $P$, a fixed policy $\pi$, and deterministic nuisance learners $\{\widehat{\eta}_N\}_{N \geq 1}$ with the following properties. For every split $N = m + n$, the propensity learner is exact, and the outcome learner satisfies*

$$\max_a \|\widehat{\mu}_{N,S,a} - \mu_{0,a}\|_{L_2(P_X)} \leq N^{-2}$$

*uniformly over training samples $S$. The population DR value is unbiased: $P\psi_\pi(\widehat{\eta}_N(S)) = V_P(\pi)$. Yet there are adjacent datasets $D \sim D'$ with the same scoring block such that*

$$|U_{\mathrm{ls}}(D, \pi) - U_{\mathrm{ls}}(D', \pi)| = n. \tag{7}$$

*Hence the global sensitivity is at least $n$ in this instance. For any fixed calibration $\Delta_0 > 0$ and privacy budget $\varepsilon > 0$, a two-policy version has bounded realized scores and the same favorable DR properties, while the exponential mechanism calibrated at $\Delta_0$ fails $\varepsilon$-DP whenever $n > 2\Delta_0$.*

The proof appears in Appendix C. It changes nuisance predictions on a region whose probability under $P_X$ vanishes with $N$. The change is negligible in the population $L_2$ norm and in the DR remainder, yet it can affect every record in a worst-case scoring block. This is the nuisance-fitting contribution quantified by Lemma 1.

The theorem rules out a plausible implication. Accurate nuisance estimation, zero population bias, and bounded realized scores do not imply a sublinear global-sensitivity bound for a learned DR utility. A valid private selection mechanism therefore needs an explicit stability bound for the nuisance learner. Appendix C also gives the two-policy likelihood-ratio argument, and Appendix F provides a nearest-neighbor construction that does not use a sample-dependent rare region.

## 4.3 Private Selection under a Deterministic Stability Bound

The exponential mechanism requires a deterministic upper bound on the movement of the realized utility. Lemma 1 shows that a learned DR utility has a direct scoring-record term and a training-record term transmitted through the nuisance predictions. We therefore impose a public uniform replace-one prediction-stability bound.

Assume the nuisance learner is deterministic, including its tie-breaking rule, and that any public randomness is fixed before the private data are observed. For every adjacent pair of fitting samples $S \sim S'$ of size $m$, suppose that

$$\max_a \sup_x |\widehat{\mu}_{S,a}(x) - \widehat{\mu}_{S',a}(x)| \leq \beta_{\mu,m}, \qquad \max_a \sup_x |\widehat{e}_{S,a}(x) - \widehat{e}_{S',a}(x)| \leq \beta_{e,m}. \tag{8}$$

We refer to (8) as the public uniform replace-one prediction-stability bound. Its constants hold for all adjacent fitting samples, are fixed independently of the private scoring block, and do not describe average-case prediction error. The induced quantity $\rho_m^\star$ below is the deterministic prediction-movement radius used in the utility-sensitivity bound.

Define

$$\rho_m^\star := (1 + \zeta^{-1})\beta_{\mu,m} + \zeta^{-2}R_{Y,\mu}\beta_{e,m}, \qquad \Delta_{\text{cert}} := \max\{2B_\psi, n\rho_m^\star\}. \tag{9}$$

Lemma 1 implies that $\Delta_{\text{cert}}$ is a global-sensitivity bound for $U_{\text{ls}}$. The mechanism fits $\widehat{\eta}$ on $D^{\text{tr}}$, evaluates each candidate on $D^{\text{sc}}$, and releases

$$\widehat{\pi} \sim \Pr(\pi) \propto \exp\left\{\frac{\varepsilon U_{\text{ls}}(D, \pi)}{2\Delta_{\text{cert}}}\right\}, \qquad \pi \in \Pi_n.$$

The stability bound calibrates privacy, while the DR remainder controls statistical bias. Let $M_n = |\Pi_n|$ and define the finite-library approximation error

$$\mathcal{A}_n(P; \Pi, \Pi_n) := \sup_{\pi \in \Pi} V_P(\pi) - \max_{\pi \in \Pi_n} V_P(\pi).$$

The next theorem states the privacy and statistical guarantees separately.

**Theorem 2** (Privacy and statistical utility). *Under the causal identification, bounded-range, public-split, data-independent-library, and deterministic stability assumptions above:*

*(i) **Privacy.** The resulting selection mechanism is pure $\varepsilon$-DP for all records in $D^{\text{tr}} \cup D^{\text{sc}}$.*

*(ii) **Statistical utility.** Suppose that, with probability at least $1 - \alpha_\eta$ over $D^{\text{tr}}$,*

$$\max_a \|\widehat{\mu}_a - \mu_{0,a}\|_{L_2(P_X)} \le r_{\mu,m}, \qquad \max_a \|\widehat{e}_a - e_{0,a}\|_{L_2(P_X)} \le r_{e,m}. \tag{10}$$

*Then, with probability at least $1 - \alpha_\eta - \alpha - \beta$,*

$$V_P(\pi_P^\star) - V_P(\widehat{\pi}) \le \mathcal{A}_n(P; \Pi, \Pi_n) + CB_\psi\sqrt{\frac{\log(2M_n/\alpha)}{n}} + C'K\zeta^{-1}r_{\mu,m}r_{e,m}$$
$$+ \frac{C''\Delta_{\text{cert}}}{n\varepsilon}\{\log M_n + \log(1/\beta)\}, \tag{11}$$

*where $C, C', C''$ are universal constants.*

The proof is in Appendix D. Part (i) is deterministic and unconditional. The probabilities in part (ii) concern nuisance estimation, concentration, and the exponential-mechanism utility guarantee.

The four terms in (11) have distinct roles. The first is the cost of restricting the release to $\Pi_n$. The second is scoring-sample concentration. The third is the DR product remainder. The final term is the cost of private selection, and it is the only term that depends on the sensitivity calibration. Since

$$\frac{\Delta_{\text{cert}}}{n\varepsilon} = \max\left\{\frac{2B_\psi}{n\varepsilon}, \frac{\rho_m^\star}{\varepsilon}\right\}, \tag{12}$$

the fixed-score contribution scales as $1/(n\varepsilon)$, whereas the nuisance-fitting contribution is governed by the deterministic score-movement bound $\rho_m^\star$. Consistency requires the approximation, concentration, DR, and privacy terms to vanish under their respective assumptions.

**Corollary 1.** *Fix $\varepsilon > 0$. If $\mathcal{A}_n(P; \Pi, \Pi_n) \to 0$, $\log(M_n)/n \to 0$, $Kr_{\mu,m}r_{e,m} \to 0$, and $\frac{\rho_m^\star}{\varepsilon}\{\log M_n + \log(1/\beta)\} \to 0$, then the regret bound in Theorem 2 vanishes for fixed failure probabilities. In particular, if a strongly convex regularized ERM learner satisfies the bound with $\rho_m^\star = O((m\lambda_m)^{-1})$, then the nuisance-fitting privacy term vanishes whenever $\frac{\log M_n + \log(1/\beta)}{\varepsilon m \lambda_m} \to 0$, while the DR statistical term requires $r_{\mu,m}r_{e,m} \to 0$.*

The proof of Corollary 1 is stated in Appendix D.

**Scope of the policy class.** The finite public library lets the analysis isolate sensitivity from nuisance fitting and yields the counting-measure exponential mechanism and the $\log M_n$ terms in the utility bound. Under the present finite-action range assumptions, the nuisance-induced sensitivity bound is already uniform over deterministic policies. An infinite or continuous class would instead require a specified base measure for the exponential mechanism, control of the measure of near-optimal policies, and uniform empirical-process bounds—for example through covering, entropy, or Rademacher-complexity conditions—to replace the finite-library counting argument. If the class is learned from protected data, policy construction is an additional private learning stage. These extensions are outside the present theorem.

**Scope of the stability bound.** The bound in (8) is sufficient and need not be necessary. It can also be loose because it must hold for every adjacent fitting sample and every prediction point. Theorem 2 does not apply to flexible learners without such a deterministic bound; sampled adjacent-pair movement remains an empirical diagnostic.

Such public uniform prediction-stability bounds are available for several deterministic strongly convex ERM learners. They follow directly from the optimization problem defining the nuisance learner.

**Remark 1.** *For deterministic finite-dimensional strongly convex regularized ERM nuisance learners with bounded features, globally $G$-Lipschitz losses, $L_g$-Lipschitz prediction maps, deterministic tie-breaking, and deterministic clipping or simplex projection,*

$$\sup_x |g_{\widehat{\theta}_S}(x) - g_{\widehat{\theta}_{S'}}(x)| \leq \frac{2L_g G}{m \lambda_m} \qquad \text{for every } S \sim S'. \tag{13}$$

*Hence $\beta_{\mu,m}$ and $\beta_{e,m}$ are explicit for this class, giving $\rho_m^\star = O((m\lambda_m)^{-1})$. Appendix E gives the multi-action statement and proof. More adaptive learners require their own deterministic stability analysis before the pure-DP guarantee applies.*

# 5 Empirical Sensitivity Audits and an ERM Positive Control

The boosted-tree experiments measure learned-utility movement on sampled adjacent datasets, while the ERM positive control uses public deterministic stability constants to instantiate the sufficient condition in Section 4.3. We examine whether favorable DR diagnostics can coexist with growing movement, whether the fixed-score scale fails on an adjacent pair, and whether regularization changes prediction and utility movement in the direction predicted by the decomposition.

## 5.1 Experimental Setup

**Data and semi-synthetic design.** A central challenge in empirical causal inference is that counterfactual outcomes are unobserved. Following established semi-synthetic practice (Curth & Van der Schaar, 2021; Curth & Van Der Schaar, 2023; Huang et al., 2024), we use covariates from the ACIC 2016 benchmark (Dorie et al., 2019) and generate treatments and outcomes from known response surfaces. After preprocessing, the design matrix has 4,802 rows and 58 encoded covariates.

The main separation audit uses a rare-region DGP, a finite-sample analogue of Theorem 1. The purpose is to create a low-probability covariate region that is nearly invisible to population-average nuisance error but can still induce large realized movement on stress scoring blocks. Let $R(x)$ indicate that a nonlinear rare-region score exceeds its empirical 95th percentile. We define

$$
\begin{aligned}
e_0(x) &= \text{clip}_{[\zeta, 1-\zeta]}\{\sigma(g_e(x) + 1.7R(x))\}, \qquad \zeta = 0.10, \\
\tau(x) &= 0.20\tanh\{1.4g_\tau(x)\} + 0.11 \cdot \mathbf{1}\{x_1 > 0\} - 0.07 \cdot \mathbf{1}\{x_6 > 0\} + 0.45R(x), \\
\mu_{0,0}(x) &= \text{clip}_{[0.02, 0.98]}\{0.25 + 0.45\sigma(g_0(x))\}, \quad \mu_{0,1}(x) = \text{clip}_{[0.02, 0.98]}\{\mu_{0,0}(x) + \tau(x)\},
\end{aligned}
$$

where $g_e, g_0, g_\tau$ are fixed sparse functions with nonlinear terms. We sample $A \sim \text{Bernoulli}(e_0(X))$ and $Y = \text{clip}_{[0,1]}\{\mu_{0,A}(X) + \xi\}$ with Gaussian noise $\xi$. The rare region affects both treatment assignment and treatment effects, while clipping enforces the overlap and bounded-outcome conditions used in the theory.

Table 1: Sampled adjacent-pair separation audit over 100 repetitions. Appendix G.2 states column definitions.

| $n$ | nuis. $L_2$ | DR prod. | unstable move/floor | naive $\log\mathrm{LR}/\varepsilon$ | audit-cal. $\log\mathrm{LR}/\varepsilon$ |
|------|--------|---------|------|------|-------|
| 300 | 0.0950 | 0.00963 | 0.37 | 0.13 | 0.053 |
| 700 | 0.0946 | 0.00958 | 0.87 | 0.37 | 0.148 |
| 1200 | 0.0952 | 0.00968 | 1.49 | 0.69 | 0.221 |
| 2000 | 0.0961 | 0.00986 | 2.40 | 1.17 | 0.319 |
| 3000 | 0.0945 | 0.00955 | 3.36 | 1.48 | 0.336 |

**Policy library, learners, and diagnostics.** The public policy library contains 160 policies: structured threshold and rare-region rules, together with linear threshold rules generated from public randomness fixed before the private data are observed. Conditional on that public randomness, the realized library is deterministic. Each repetition uses $m = 1500$ nuisance-fitting records and $n = 1500$ policy-scoring records unless stated otherwise. True policy values are approximated by Monte Carlo evaluation of the known conditional means, and regret for randomized selection mechanisms is reported in expectation under the selection distribution. Propensities are fit by logistic regression and outcome regressions by fixed-seed boosted trees. We compare a regularized default variant with a deeper, weakly regularized stress variant. Under the imposed overlap $\zeta = 0.10$, the bounded-range construction gives $B_\psi = 1 + 1/\zeta = 11$, so the fixed-score sensitivity floor is $2B_\psi = 22$. For adjacent datasets $D, D'$, we audit learned-utility movement by $\widehat{\Delta}_{\mathrm{move}} := \max_{\pi \in \Pi_n} |U_{\mathrm{ls}}(D, \pi) - U_{\mathrm{ls}}(D', \pi)|$. We also report the DR product proxy $\widehat{r}_{\mathrm{DR}} := \frac{1}{2}\{\mathrm{RMSE}(\widehat{\mu}_0) + \mathrm{RMSE}(\widehat{\mu}_1)\}\mathrm{RMSE}(\widehat{e})$, and, in the regularization and split experiments, the prediction-movement proxy $\widehat{\rho}_{\mathrm{pred}}$ from Lemma 1.

Appendix G.1 gives more specific details of preprocessing, DGP, stress-audit construction, policy construction, learner hyperparameters, experiment grid, and remaining diagnostics.

## 5.2 Results and Analysis

**DR accuracy can coexist with growing audited movement.** We construct sampled adjacent nuisance-fitting datasets that differ in one rare-region anchor record and evaluate the two learned utilities on scoring blocks of increasing size. Table 1 reports averages over 100 repetitions. Nuisance $L_2$ error and the DR product proxy remain nearly constant as $n$ grows. The observed utility movement rises from 0.37 to 3.36 times the fixed-score floor $2B_\psi$. This pattern is consistent with the $n\rho_m$ term in Lemma 1: one fitting-record change alters the learned score map, and the scoring sum accumulates that change.

Figure 1 evaluates the same sampled adjacent pairs. The fixed-score scale $\Delta = 2B_\psi$ is valid when the score map is fixed or frozen. For the unstable learned score, the observed movement eventually exceeds this scale. The empirical maximum log-likelihood ratio divided by $\varepsilon$ reaches 1.17 at $n = 2000$ and 1.48 at $n = 3000$, so the fixed-score calibration violates the target inequality on these audited pairs. The audit-calibrated scale is constructed from the same observed movements. Its subunit likelihood-ratio values are therefore an in-sample implementation check rather than independent validation.

**Regularization reduces observed prediction and utility movement.** Because XGBoost is outside the strongly convex ERM class covered in Appendix E, Table 2 reports empirical prediction-movement proxies. As $\lambda$ increases, the sampled prediction-movement proxy falls from 8.87 to 0.48, and the observed utility movement falls from 40.18 to 5.15, supporting the qualitative direction of the sensitivity decomposition.

Appendix G.3 compares the sampled audit scale with the scale obtained by inserting the empirical prediction proxy into the theorem's formula. The latter remains much larger. Appendix G.4 then replaces the boosted-tree nuisances with a strongly convex ERM learner whose stability constants are public and deterministic. In this positive control, the resulting prediction-movement radius is 26.87 times the sampled proxy, which covers all audited movements but is conservative.

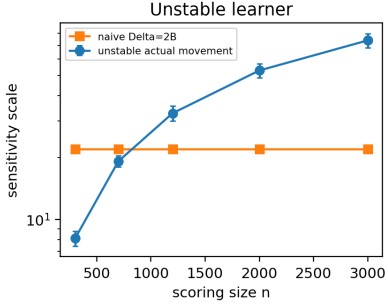 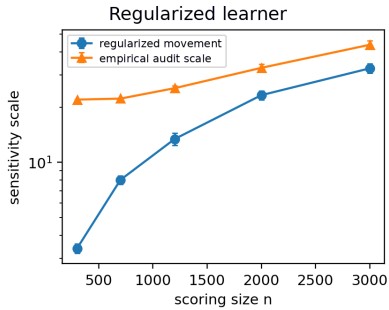 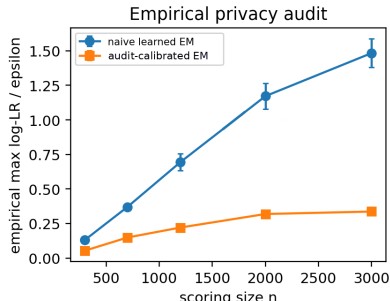

Figure 1: Empirical adjacent-pair audits across scoring sizes. Left: learned-score movement for the unstable learner versus the fixed-score floor $\Delta = 2B_\psi$. Middle: movement for the regularized learner versus the empirical audit scale. Right: the maximum audited log-likelihood ratio divided by $\varepsilon$. A value above one witnesses failure of the proposed calibration on an audited pair.

Table 2: Regularization diagnostic over 100 repetitions. Appendix G.2 states column definitions.

| $\lambda$ | $\widehat{\rho}_{\mathrm{pred}}$ | move | audit $\Delta$ | audit/floor | floored $\Delta$ | floor rate | regret |
|------|------|-------|-------|------|-------|------|--------|
| 0.01 | 8.87 | 40.18 | 52.23 | 2.37 | 54.70 | 0.23 | 0.0211 |
| 0.03 | 8.84 | 39.72 | 51.63 | 2.35 | 54.13 | 0.26 | 0.0210 |
| 0.10 | 8.72 | 39.27 | 51.05 | 2.32 | 53.55 | 0.23 | 0.0209 |
| 0.30 | 8.45 | 37.39 | 48.60 | 2.21 | 51.11 | 0.24 | 0.0207 |
| 1.00 | 7.60 | 33.94 | 44.12 | 2.01 | 46.55 | 0.25 | 0.0201 |
| 3.00 | 4.63 | 25.41 | 33.04 | 1.50 | 36.15 | 0.41 | 0.0185 |
| 10.0 | 2.37 | 17.54 | 22.80 | 1.04 | 27.76 | 0.57 | 0.0167 |
| 30.0 | 1.39 | 12.94 | 16.82 | 0.76 | 23.90 | 0.77 | 0.0157 |
| 100  | 0.48 | 5.15  | 6.70  | 0.30 | 22.00 | 1.00 | 0.0149 |

**Sample-splitting decomposition.** Holding the public policy library fixed, the split experiment isolates three sample-dependent terms in the regret bound: nuisance quality, scoring-sample concentration, and the learned-score movement component. We fix the total sample size at 3,600 and vary the nuisance-fitting fraction. Table 3 shows the resulting trade-off. Increasing the fitting fraction from 0.20 to 0.80 improves nuisance RMSE from 0.0838 to 0.0608 and reduces observed movement from 2.61 to 0.51, reflecting the stability side of the bound. At the same time, the scoring block shrinks from 2,880 to 720 records, $1/\sqrt{n}$ increases, and empirical expected regret rises from 0.0232 to 0.0326, reflecting the concentration side of the bound. The audit scale after applying the fixed-score floor remains 22 for every split because the observed adjacent-pair movements are below $2B_\psi = 22$.

In summary, the sampled separation audit exhibits favorable DR diagnostics together with increasing utility movement and a fixed-score calibration failure on specific adjacent pairs. The regularization audit shows that observed prediction movement and utility movement decline together, and the sample-split experiment illustrates the trade-off among nuisance accuracy, scoring concentration, and observed movement. The ERM positive control in Appendix G.4 complements these diagnostics with a public deterministic stability bound that instantiates Theorem 2.

# 6 Conclusion

Private causal-learning methods place the privacy mechanism at different stages. Some use DP-compatible nuisance or base learners. Others retain non-private nuisances and protect downstream predictions, aggregation, or a specified causal estimand. We analyzed direct private policy selection from a finite public library, where the released object is one policy identity and the exponential mechanism operates on nuisance-fitted comparison utilities.

Table 3: Sample-split decomposition over 100 repetitions. Appendix G.2 states column definitions.

| fit frac. | $m$ | $n$ | nuis. RMSE | $\widehat{\rho}_{\mathrm{pred}}$ | move | floored $\Delta$ | floor rate | $1/\sqrt{n}$ | regret |
|---|---|---|---|---|---|---|---|---|---|
| 0.20 | 720 | 2880 | 0.0838 | 9.04 | 2.61 | 22.00 | 1.00 | 0.0186 | 0.0232 |
| 0.25 | 900 | 2700 | 0.0785 | 6.61 | 2.12 | 22.00 | 1.00 | 0.0192 | 0.0242 |
| 0.35 | 1260 | 2340 | 0.0725 | 4.61 | 1.41 | 22.00 | 1.00 | 0.0207 | 0.0258 |
| 0.50 | 1800 | 1800 | 0.0664 | 2.93 | 1.01 | 22.00 | 1.00 | 0.0236 | 0.0284 |
| 0.65 | 2340 | 1260 | 0.0631 | 2.69 | 0.73 | 22.00 | 1.00 | 0.0282 | 0.0304 |
| 0.75 | 2700 | 900 | 0.0615 | 1.89 | 0.56 | 22.00 | 1.00 | 0.0333 | 0.0320 |
| 0.80 | 2880 | 720 | 0.0608 | 1.80 | 0.51 | 22.00 | 1.00 | 0.0373 | 0.0326 |

For a fixed score map, bounded summands yield constant global sensitivity. When the score map is learned on the protected training block, one replacement can change every score in the scoring block. The resulting nuisance-fitting contribution can be linear in the scoring-sample size even when the propensity model is exact, the outcome-regression error vanishes, the population DR value is unbiased, and the usual orthogonal product remainder is controlled. These statistical properties therefore do not determine the worst-case sensitivity of the realized learned utility.

A public uniform replace-one prediction-stability bound supplies one sufficient calibration. It gives an unconditional pure-$\varepsilon$-DP guarantee and a separate high-probability regret result, but it is learner-specific and can be conservative. The boosted-tree experiments provide sampled sensitivity diagnostics, while the ERM positive control illustrates the calibration under public deterministic stability constants.

Private nuisance training, downstream private aggregation, joint private value or function release, private ERM, and valid smooth-sensitivity mechanisms protect different objects or stages of the pipeline. For direct private policy selection, the complete learned utility requires an explicit sensitivity bound. Extending the result to a continuous class would additionally require a base measure and near-optimal-set control for the exponential mechanism, together with uniform statistical complexity bounds. Constructing the class from protected data would add a private learning stage.

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

## Appendix

## A  Fixed and Frozen Utilities

**Proposition 1.** *If $|q_\pi(z)| \leq B$ for all $\pi, z$, then $\Delta_{\text{fix}} \leq 2B$. If $\eta^\dagger$ is fixed relative to the protected scoring block and $|\psi_\pi(z; \eta^\dagger)| \leq B_\psi$ for all $\pi, z$, then under scoring-block adjacency with $\eta^\dagger$ held fixed, $\Delta_{\text{fr}} \leq 2B_\psi$.*

The proof of Proposition 1 is stated below.

*Proof of Proposition 1.* Fix an arbitrary policy $\pi \in \Pi_n$. If $D \sim D'$ differ in one scoring record, then all summands except the replaced record coincide. Writing the replaced records as $Z$ and $Z'$, we have

$$|U_{\text{fix}}(D, \pi) - U_{\text{fix}}(D', \pi)| = |q_\pi(Z) - q_\pi(Z')| \leq 2B.$$

If the replacement occurs outside the scoring block and the score map is fixed, every summand is identical and the utility change is zero. Taking the supremum over $\pi$ proves the fixed-score bound.

For frozen DR scores, condition on the fixed value of $\eta^\dagger$. Under the stated adjacency relation, replacing a protected user does not alter $\eta^\dagger$, so the function $z \mapsto \psi_\pi(z; \eta^\dagger)$ is a deterministic bounded score map. If the replacement is in the scoring/release block, the same one-summand calculation gives

$$|U_{\text{fr}}(D, \pi) - U_{\text{fr}}(D', \pi)| \leq 2B_\psi.$$

If the replacement is outside the scoring/release block and leaves $\eta^\dagger$ fixed, the frozen utility is unchanged. Taking the supremum over $\pi$ gives $\Delta_{\text{fr}} \leq 2B_\psi$. The proof therefore protects only records whose replacement is covered by this adjacency relation. Auxiliary records used to build $\eta^\dagger$ are not protected by this argument unless their data dependence is separately privatized or included in the sensitivity calculation. $\qquad\square$

## B  DR Score Lipschitz Bound and Sensitivity Decomposition

**Lemma 2.** *Let $\eta = (\mu, e)$ and $\eta' = (\mu', e')$ satisfy the deterministic ranges. If*

$$\max_a \sup_x |\mu_a(x) - \mu'_a(x)| \leq \delta_\mu, \qquad \max_a \sup_x |e_a(x) - e'_a(x)| \leq \delta_e,$$

*then for every $z$ and $\pi$,*
$$|\psi_\pi(z; \eta) - \psi_\pi(z; \eta')| \leq (1 + \zeta^{-1})\delta_\mu + \zeta^{-2} R_{Y,\mu} \delta_e.$$

The proof of Lemma 2 is stated below.

*Proof of Lemma 2.* Write $z = (x, A, Y)$, set $a = \pi(x)$, and let $I = \mathbf{1}\{A = a\}$. From (1),

$$\psi_\pi(z; \eta) - \psi_\pi(z; \eta') = \{\mu_a(x) - \mu'_a(x)\} + I\left\{\frac{Y - \mu_a(x)}{e_a(x)} - \frac{Y - \mu'_a(x)}{e'_a(x)}\right\}.$$

For the inverse-propensity part, add and subtract $(Y - \mu'_a(x))/e_a(x)$ to obtain

$$\left|\frac{Y - \mu_a(x)}{e_a(x)} - \frac{Y - \mu'_a(x)}{e'_a(x)}\right| \leq \frac{|\mu_a(x) - \mu'_a(x)|}{e_a(x)} + |Y - \mu'_a(x)|\left|\frac{1}{e_a(x)} - \frac{1}{e'_a(x)}\right|.$$

The deterministic range assumptions give $e_a(x), e'_a(x) \geq \zeta$ and $|\mu_a(x) - \mu'_a(x)| \leq \delta_\mu$. They also give $|Y - \mu'_a(x)| \leq R_{Y,\mu}$: since $Y \in [\underline{y}, \overline{y}]$ and $\mu'_a(x) \in [\underline{\mu}, \overline{\mu}]$, the maximum of $|y - u|$ over this rectangle is $\max\{|\overline{y} - \underline{\mu}|, |\underline{y} - \overline{\mu}|\}$. Finally,

$$\left|\frac{1}{e_a(x)} - \frac{1}{e'_a(x)}\right| = \frac{|e_a(x) - e'_a(x)|}{e_a(x)e'_a(x)} \leq \zeta^{-2}\delta_e.$$

Combining these bounds and using $I \leq 1$ yields

$$|\psi_\pi(z;\eta) - \psi_\pi(z;\eta')| \leq \delta_\mu + \left\{\zeta^{-1}\delta_\mu + R_{Y,\mu}\zeta^{-2}\delta_e\right\}$$
$$= (1 + \zeta^{-1})\delta_\mu + \zeta^{-2}R_{Y,\mu}\delta_e.$$

$\square$

*Proof of Lemma 1.* If adjacent datasets differ in one scoring record, $D^{\mathrm{tr}}$ is unchanged. Hence $\widehat{\eta}(D^{\mathrm{tr}})$ is fixed and only one bounded score changes, giving

$$|U_{\mathrm{ls}}(D,\pi) - U_{\mathrm{ls}}(D',\pi)| \leq 2B_\psi.$$

If they differ in one training record, write the adjacent training samples as $S \sim S'$. The scoring block is unchanged, while the nuisance changes from $\widehat{\eta}_S$ to $\widehat{\eta}_{S'}$. For every fixed $\pi$,

$$|U_{\mathrm{ls}}(D,\pi) - U_{\mathrm{ls}}(D',\pi)|$$
$$\leq \sum_{i \in D^{\mathrm{sc}}} |\psi_\pi\{Z_i;\widehat{\eta}_S\} - \psi_\pi\{Z_i;\widehat{\eta}_{S'}\}|$$
$$\leq n\left\{(1 + \zeta^{-1})\Gamma_{\mu,m} + \zeta^{-2}R_{Y,\mu}\Gamma_{e,m}\right\} = n\rho_m,$$

where the second inequality is Lemma 2 together with the definitions of $\Gamma_{\mu,m}$ and $\Gamma_{e,m}$. Taking the supremum over $\pi$ and the maximum over the two possible replacement locations proves the result. $\square$

## C   Proof of Theorem 1: Separation and Naive Calibration Violation

This appendix proves the separation theorem. It first constructs the learned-utility sensitivity example whose change is proportional to the scoring-sample size, and then gives the two-policy likelihood-ratio argument showing that a fixed-score calibration can violate pure DP.

### C.1   Sensitivity Proportional to the Scoring-Sample Size

*Proof of the separation part of Theorem 1.* We construct a single population distribution and a triangular sequence of deterministic learners. Let $P$ be the distribution with binary actions $\mathcal{A} = \{1,2\}$, covariates $X \sim \mathrm{Unif}[0,1]$, potential outcomes $Y(1) = Y(2) = 0$, and propensities $e_{0,1}(x) = e_{0,2}(x) = 1/2$ for all $x$. Let the policy be $\pi_1(x) \equiv 1$. The true outcome nuisances are $\mu_{0,1} = \mu_{0,2} = 0$, and $V_P(\pi_1) = 0$.

For each sample size $N = m + n$, define the rare sets

$$B_N = [0, N^{-4}], \qquad C_N = (N^{-4}, 2N^{-4}],$$

and define the deterministic trigger functional

$$T_N(S) = \mathbf{1}\{\exists j \in S : X_j \in C_N\}.$$

The learner $\widehat{\eta}_N$ maps a nuisance-fitting sample $S$ to

$$\widehat{e}_{N,S,1}(x) = \widehat{e}_{N,S,2}(x) = 1/2,$$

$$\widehat{\mu}_{N,S,1}(x) = \mathbf{1}\{x \in B_N\}T_N(S), \qquad \widehat{\mu}_{N,S,2}(x) = 0.$$

This definition is allowed to depend on $N$, as stated in the theorem, but the population distribution $P$ is fixed across $N$.

For every training sample $S$,

$$\max_a \|\widehat{\mu}_{N,S,a} - \mu_{0,a}\|_{L_2(P_X)} \leq \|\mathbf{1}\{\cdot \in B_N\}\|_{L_2(P_X)} = \sqrt{P_X(B_N)} = N^{-2},$$

and $\max_a \|\widehat{e}_{N,S,a} - e_{0,a}\|_{L_2(P_X)} = 0$. Because the propensity nuisance is exactly correct, the product identity (2) gives

$$P\psi_{\pi_1}(\widehat{\eta}_N(S)) - V_P(\pi_1) = 0$$

for every training sample $S$.

It remains to show large worst-case sensitivity. Construct adjacent training blocks $S \sim S'$ as follows. Let $S$ contain exactly one record with $X \in C_N$, let $S'$ be obtained by replacing that record with a record whose covariate is outside $C_N$, and let all other training covariates in both samples lie outside $C_N$. Then $T_N(S) = 1$ and $T_N(S') = 0$. Use a common scoring block for $D = (S, D^{\mathrm{sc}})$ and $D' = (S', D^{\mathrm{sc}})$ with every scoring record satisfying $X_i \in B_N$, $A_i = 2$, and $Y_i = 0$. These records have vanishing probability under $P^n$, but they are valid points in the sample space. Pure-DP global sensitivity is worst-case over all adjacent datasets in the sample space, not high-probability under $P^n$.

For each scoring record in this block, $A_i \neq \pi_1(X_i)$, so the inverse-propensity correction in (1) is zero and

$$\psi_{\pi_1}\{Z_i; \widehat{\eta}_N(S)\} = \widehat{\mu}_{N,S,1}(X_i) = 1, \qquad \psi_{\pi_1}\{Z_i; \widehat{\eta}_N(S')\} = \widehat{\mu}_{N,S',1}(X_i) = 0.$$

Therefore each of the $n$ scoring summands changes by one, and

$$|U_{\mathrm{ls}}(D, \pi_1) - U_{\mathrm{ls}}(D', \pi_1)| = n.$$

This proves the sensitivity scaling with the scoring-sample size, correct-propensity, nuisance-accuracy, and zero-bias claims for the single-policy construction. $\qquad\square$

### C.2 Two-Policy DP Violation Under Naive Calibration

*Proof of the two-policy part of Theorem 1.* Use the same fixed population distribution as in Appendix C: $X \sim \mathrm{Unif}[0,1]$, $\mathcal{A} = \{1,2\}$, $Y(1) = Y(2) = 0$, and $e_{0,1} = e_{0,2} = 1/2$. Let $\Pi_n = \{\pi_0, \pi_1\}$ with $\pi_0(x) \equiv 2$ and $\pi_1(x) \equiv 1$. For $N = m + n$, let

$$B_N = [0, N^{-4}], \qquad C_N = (N^{-4}, 2N^{-4}], \qquad T_N(S) = \mathbf{1}\{\exists j \in S : X_j \in C_N\}.$$

Define the learner, for every training sample $S$, by

$$\widehat{e}_{N,S,1}(x) = \widehat{e}_{N,S,2}(x) = 1/2, \qquad \widehat{\mu}_{N,S,2}(x) = 0,$$

and

$$\widehat{\mu}_{N,S,1}(x) = \mathbf{1}\{x \in B_N\}\{2T_N(S) - 1\}.$$

This is a fully specified deterministic sample-size-dependent nuisance learner.

We verify the statistical properties first. The true outcome nuisances are zero, so for every $S$,

$$\max_a \|\widehat{\mu}_{N,S,a} - \mu_{0,a}\|_{L_2(P_X)} = \|\mathbf{1}\{\cdot \in B_N\}\|_{L_2(P_X)} = N^{-2},$$

and the propensity error is zero. Since $\widehat{e}_N = e_0$, Lemma 3 gives

$$P\psi_{\pi_b}(\widehat{\eta}_N(S)) = V_P(\pi_b) = 0, \qquad b \in \{0,1\},$$

for every training sample $S$. The realized scores are uniformly bounded. Indeed, for $\pi_0$, $\mu_2 \equiv 0$ and $Y = 0$, so $\psi_{\pi_0} \equiv 0$. For $\pi_1$, outside $B_N$ the learned mean is zero and the score is zero. On $B_N$, $\widehat{\mu}_1$ equals either $+1$ or $-1$. If $A = 2$, the inverse-propensity term is absent and the score is $\pm 1$, while if $A = 1$ the score is

$$\widehat{\mu}_1(X) + 2\{0 - \widehat{\mu}_1(X)\} = -\widehat{\mu}_1(X),$$

also in $\{-1, +1\}$. Hence $|\psi_{\pi_b}(z; \widehat{\eta}_N(S))| \leq 1$ for $b = 0, 1$, all $z$, and all $S$.

Now choose adjacent training blocks $S \sim S'$ such that $T_N(S) = 1$ and $T_N(S') = 0$, as in Appendix C. Use the same scoring block in both datasets, with $X_i \in B_N$, $A_i = 2$, and $Y_i = 0$ for all $i = 1, \ldots, n$. Then

$$U_{\mathrm{ls}}(D, \pi_0) = U_{\mathrm{ls}}(D', \pi_0) = 0, \qquad U_{\mathrm{ls}}(D, \pi_1) = n, \qquad U_{\mathrm{ls}}(D', \pi_1) = -n.$$

For the exponential mechanism calibrated at the fixed value $\Delta_0$, put $a = \varepsilon n/(2\Delta_0)$. The probability of outputting $\pi_1$ under $D$ is $e^a/(1 + e^a)$, while under $D'$ it is $e^{-a}/(1 + e^{-a}) = 1/(1 + e^a)$. Therefore

$$\frac{\mathbb{P}\{\mathcal{M}(D) = \pi_1\}}{\mathbb{P}\{\mathcal{M}(D') = \pi_1\}} = e^a = \exp\left\{\frac{\varepsilon n}{2\Delta_0}\right\}.$$

If $n > 2\Delta_0$, this ratio is strictly larger than $e^\varepsilon$. Taking the measurable event $\{\pi_1\}$ violates the defining likelihood-ratio inequality for pure $\varepsilon$-DP. $\qquad\square$

## D  Proofs of Theorem 2 and Corollary 1

This appendix proves the learned-score selection result in three steps. First, Appendix D.1 proves the pure-DP claim using the stability bound from Section 4.3. Second, Lemma 4 proves uniform statistical accuracy of the learned DR utilities by combining scoring-sample concentration with the DR product remainder. Third, Lemma 5 gives the finite-library exponential-mechanism oracle inequality, after which we assemble the regret bound in Theorem 2. Corollary 1 is then an immediate consequence of the four vanishing terms in the theorem.

### D.1  Differential Privacy Guarantee

*Proof of the DP part of Theorem 2.* The stability bound gives $\Gamma_{\mu,m} \leq \beta_{\mu,m}$ and $\Gamma_{e,m} \leq \beta_{e,m}$, hence Lemma 1 gives

$$\sup_{D \sim D'} \sup_{\pi \in \Pi_n} |U_{\mathrm{ls}}(D, \pi) - U_{\mathrm{ls}}(D', \pi)| \leq \Delta_{\mathrm{cert}}.$$

The standard exponential-mechanism proof then applies. The sensitivity bound implies both $U_{\mathrm{ls}}(D, \pi) \leq U_{\mathrm{ls}}(D', \pi) + \Delta_{\mathrm{cert}}$ and $U_{\mathrm{ls}}(D, \rho) \geq U_{\mathrm{ls}}(D', \rho) - \Delta_{\mathrm{cert}}$ for every $\pi, \rho \in \Pi_n$. Hence the numerator ratio is at most $\exp\{\varepsilon/2\}$, and the normalizers satisfy

$$\sum_\rho \exp\{\varepsilon U_{\mathrm{ls}}(D, \rho)/(2\Delta_{\mathrm{cert}})\} \geq e^{-\varepsilon/2} \sum_\rho \exp\{\varepsilon U_{\mathrm{ls}}(D', \rho)/(2\Delta_{\mathrm{cert}})\}.$$

Therefore, for any output $\pi$,

$$\frac{\mathbb{P}\{\widehat{\pi} = \pi \mid D\}}{\mathbb{P}\{\widehat{\pi} = \pi \mid D'\}} \leq \exp\{\varepsilon/2\} \cdot \exp\{\varepsilon/2\} = e^\varepsilon.$$

Summing over events gives pure $\varepsilon$-DP. $\qquad\square$

### D.2  Doubly Robust Identity and Product Remainder

**Lemma 3.** *For any nuisance pair $\eta = (\mu, e)$ satisfying $e_a(x) > 0$ for all $a, x$ and any deterministic policy $\pi$,*

$$P\psi_\pi(\eta) - V_P(\pi) = \sum_{a=1}^K \mathbb{E}\left[\mathbf{1}\{\pi(X) = a\}\frac{e_a(X) - e_{0,a}(X)}{e_a(X)}\{\mu_a(X) - \mu_{0,a}(X)\}\right].$$

*Consequently, under the overlap range $e_a(x) \geq \zeta$,*

$$|P\psi_\pi(\eta) - V_P(\pi)| \leq \zeta^{-1} \sum_{a=1}^K \|\mu_a - \mu_{0,a}\|_{L_2(P_X)} \|e_a - e_{0,a}\|_{L_2(P_X)}.$$

The proof of Lemma 3 is stated below.

*Proof of Lemma 3.* Fix $x$ and let $a = \pi(x)$. By consistency and unconfoundedness,

$$\mathbb{E}[Y \mid A = a, X = x] = \mu_{0,a}(x), \qquad \mathbb{P}(A = a \mid X = x) = e_{0,a}(x).$$

Taking the conditional expectation of the score in (1) given $X = x$ gives

$$
\begin{aligned}
\mathbb{E}[\psi_\pi(Z;\eta) \mid X = x] &= \mu_a(x) + \frac{e_{0,a}(x)}{e_a(x)}\{\mu_{0,a}(x) - \mu_a(x)\} \\
&= \mu_{0,a}(x) + \left(1 - \frac{e_{0,a}(x)}{e_a(x)}\right)\{\mu_a(x) - \mu_{0,a}(x)\} \\
&= \mu_{0,a}(x) + \frac{e_a(x) - e_{0,a}(x)}{e_a(x)}\{\mu_a(x) - \mu_{0,a}(x)\}.
\end{aligned}
$$

Since $V_P(\pi) = \mathbb{E}[\mu_{0,\pi(X)}(X)]$, taking expectation over $X$ and decomposing according to the events $\{\pi(X) = a\}$ proves the identity.

We now prove the product-remainder bound explicitly. Starting from the identity,

$$
\begin{aligned}
&|P\psi_\pi(\eta) - V_P(\pi)| \\
&\leq \sum_{a=1}^{K} \mathbb{E}\left[\mathbf{1}\{\pi(X) = a\}\left|\frac{e_a(X) - e_{0,a}(X)}{e_a(X)}\right||\mu_a(X) - \mu_{0,a}(X)|\right] \\
&\leq \zeta^{-1} \sum_{a=1}^{K} \mathbb{E}\left[\mathbf{1}\{\pi(X) = a\}|e_a(X) - e_{0,a}(X)||\mu_a(X) - \mu_{0,a}(X)|\right] \\
&\leq \zeta^{-1} \sum_{a=1}^{K} \mathbb{E}\left[|e_a(X) - e_{0,a}(X)||\mu_a(X) - \mu_{0,a}(X)|\right] \\
&\leq \zeta^{-1} \sum_{a=1}^{K} \left\{\mathbb{E}|e_a(X) - e_{0,a}(X)|^2\right\}^{1/2}\left\{\mathbb{E}|\mu_a(X) - \mu_{0,a}(X)|^2\right\}^{1/2} \\
&= \zeta^{-1} \sum_{a=1}^{K} \|e_a - e_{0,a}\|_{L_2(P_X)}\|\mu_a - \mu_{0,a}\|_{L_2(P_X)}.
\end{aligned}
$$

The first inequality is the triangle inequality, the second uses $e_a(X) \geq \zeta$, the third uses $\mathbf{1}\{\pi(X) = a\} \leq 1$, and the fourth is Cauchy–Schwarz applied to the $a$th summand. $\qquad\square$

### D.3 Regret Bound and Vanishing-Regret Corollary

**Lemma 4.** *Under the deterministic ranges and nuisance accuracy (10), with probability at least $1 - \alpha_\eta - \alpha$,*

$$
\sup_{\pi \in \Pi_n}\left|n^{-1}U_{\mathrm{ls}}(D, \pi) - V_P(\pi)\right| \leq CB_\psi\sqrt{\frac{\log(2M_n/\alpha)}{n}} + K\zeta^{-1}r_{\mu,m}r_{e,m}.
$$

The proof of Lemma 4 is stated below.

*Proof of Lemma 4.* Define the nuisance event

$$
\mathcal{E}_\eta = \left\{\max_a \|\widehat{\mu}_a - \mu_{0,a}\|_{L_2(P_X)} \leq r_{\mu,m}, \quad \max_a \|\widehat{e}_a - e_{0,a}\|_{L_2(P_X)} \leq r_{e,m}\right\}.
$$

By assumption, $\mathbb{P}(\mathcal{E}_\eta) \geq 1 - \alpha_\eta$ over the nuisance-fitting block. Condition on an arbitrary realization of $D^{\mathrm{tr}}$, so that $\widehat{\eta} = \widehat{\eta}(D^{\mathrm{tr}})$ is fixed. Conditional on this training block, the scoring records are independent, and for every fixed $\pi \in \Pi_n$ the variables $\psi_\pi(Z_{m+i}; \widehat{\eta})$ satisfy $|\psi_\pi(Z_{m+i}; \widehat{\eta})| \leq B_\psi$. Hence each summand lies in an interval of length at most $2B_\psi$. Hoeffding's inequality gives, for every $t > 0$,

$$
\mathbb{P}\left(\left|n^{-1}U_{\mathrm{ls}}(D, \pi) - P\psi_\pi(\widehat{\eta})\right| > t \,\middle|\, D^{\mathrm{tr}}\right) \leq 2\exp\left\{-\frac{nt^2}{2B_\psi^2}\right\}.
$$

Choose

$$t = B_\psi \sqrt{\frac{2 \log(2M_n/\alpha)}{n}}.$$

Then

$$2 \exp\left\{ -\frac{nt^2}{2B_\psi^2} \right\} = 2 \exp\{-\log(2M_n/\alpha)\} = \frac{\alpha}{M_n}.$$

Thus, for each fixed $\pi$,

$$\mathbb{P}\left( \left| n^{-1} U_{\mathrm{ls}}(D, \pi) - P\psi_\pi(\widehat{\eta}) \right| > B_\psi \sqrt{\frac{2 \log(2M_n/\alpha)}{n}} \,\Bigg|\, D^{\mathrm{tr}} \right) \le \frac{\alpha}{M_n}.$$

Applying the union bound over the $M_n$ policies,

$$\mathbb{P}\left( \sup_{\pi \in \Pi_n} \left| n^{-1} U_{\mathrm{ls}}(D, \pi) - P\psi_\pi(\widehat{\eta}) \right| > B_\psi \sqrt{\frac{2 \log(2M_n/\alpha)}{n}} \,\Bigg|\, D^{\mathrm{tr}} \right)$$

$$\le \sum_{\pi \in \Pi_n} \mathbb{P}\left( \left| n^{-1} U_{\mathrm{ls}}(D, \pi) - P\psi_\pi(\widehat{\eta}) \right| > B_\psi \sqrt{\frac{2 \log(2M_n/\alpha)}{n}} \,\Bigg|\, D^{\mathrm{tr}} \right) \le \alpha.$$

Equivalently, there is a conditional scoring event $\mathcal{E}_{\mathrm{sc}}(D^{\mathrm{tr}})$ with conditional probability at least $1-\alpha$ on which

$$\sup_{\pi \in \Pi_n} \left| n^{-1} U_{\mathrm{ls}}(D, \pi) - P\psi_\pi(\widehat{\eta}) \right| \le C B_\psi \sqrt{\frac{\log(2M_n/\alpha)}{n}},$$

where $C$ absorbs the displayed constant $\sqrt{2}$. Because the conditional bound holds for every training block, integrating over $D^{\mathrm{tr}}$ gives $\mathbb{P}(\mathcal{E}_{\mathrm{sc}}) \ge 1 - \alpha$, and hence $\mathbb{P}(\mathcal{E}_\eta \cap \mathcal{E}_{\mathrm{sc}}) \ge 1 - \alpha_\eta - \alpha$.

On $\mathcal{E}_\eta$, Lemma 3 gives, uniformly over $\pi \in \Pi_n$,

$$|P\psi_\pi(\widehat{\eta}) - V_P(\pi)| \le \sum_{a=1}^{K} \zeta^{-1} \|\widehat{e}_a - e_{0,a}\|_{L_2(P_X)} \|\widehat{\mu}_a - \mu_{0,a}\|_{L_2(P_X)} \le K\zeta^{-1} r_{\mu,m} r_{e,m}.$$

On $\mathcal{E}_\eta \cap \mathcal{E}_{\mathrm{sc}}$, the triangle inequality combines the concentration bound and the product-remainder bound to prove the stated uniform value bound. $\qquad\square$

**Lemma 5.** *Condition on any dataset $D$. If the exponential mechanism uses utility $U_{\mathrm{ls}} = n\widehat{V}_{\mathrm{ls}}$ and sensitivity $\Delta_{\mathrm{cert}}$, then with probability at least $1 - \beta$,*

$$\max_{\pi \in \Pi_n} \widehat{V}_{\mathrm{ls}}(\pi) - \widehat{V}_{\mathrm{ls}}(\widehat{\pi}) \le \frac{2\Delta_{\mathrm{cert}}}{n\varepsilon} \{\log M_n + \log(1/\beta)\}.$$

The proof of Lemma 5 is stated below.

*Proof of Lemma 5.* Let $U^* = \max_{\pi \in \Pi_n} U_{\mathrm{ls}}(D, \pi)$ and

$$t = \frac{2\Delta_{\mathrm{cert}}}{\varepsilon} \{\log M_n + \log(1/\beta)\}.$$

Define the bad set

$$\mathcal{B}_t = \{\pi \in \Pi_n : U_{\mathrm{ls}}(D, \pi) \le U^* - t\}.$$

Since at least one policy attains utility $U^*$, the exponential-mechanism normalizer is at least $\exp\{\varepsilon U^*/(2\Delta_{\mathrm{cert}})\}$. Therefore,

$$\mathbb{P}(\widehat{\pi} \in \mathcal{B}_t \mid D) \le \frac{|\mathcal{B}_t| \exp\{\varepsilon(U^* - t)/(2\Delta_{\mathrm{cert}})\}}{\exp\{\varepsilon U^*/(2\Delta_{\mathrm{cert}})\}}$$

$$\le M_n \exp\{-\varepsilon t/(2\Delta_{\mathrm{cert}})\} = \beta.$$

Thus, with conditional probability at least $1 - \beta$, the selected policy has utility loss at most $t$. Dividing by $n$ proves the stated inequality for $\widehat{V}_{\mathrm{ls}} = n^{-1} U_{\mathrm{ls}}$. $\qquad\square$

*Proof of the Regret Bound in Theorem 2.* The pure-DP claim was proved in Appendix D.1. We now prove the regret inequality with the advertised probability. Let

$$\widehat{V}_{\mathrm{ls}}(\pi) = n^{-1} U_{\mathrm{ls}}(D, \pi), \qquad \pi_n^\star \in \arg \max_{\pi \in \Pi_n} V_P(\pi).$$

Let $\mathcal{E}_{\mathrm{val}}$ be the event from Lemma 4. It has probability at least $1 - \alpha_\eta - \alpha$ over $D^{\mathrm{tr}}, D^{\mathrm{sc}}$. Conditional on the realized dataset $D$, Lemma 5 gives an event $\mathcal{E}_{\mathrm{em}}(D)$ over only the exponential-mechanism randomness with conditional probability at least $1 - \beta$ on which

$$\max_{\pi \in \Pi_n} \widehat{V}_{\mathrm{ls}}(\pi) - \widehat{V}_{\mathrm{ls}}(\widehat{\pi}) \leq \frac{2\Delta_{\mathrm{cert}}}{n\varepsilon} \{\log M_n + \log(1/\beta)\}.$$

Integrating this conditional probability bound over the data and applying a union bound gives

$$\mathbb{P}(\mathcal{E}_{\mathrm{val}} \cap \mathcal{E}_{\mathrm{em}}(D)) \geq 1 - \alpha_\eta - \alpha - \beta.$$

On this intersection of events,

$$\begin{aligned}
V_P(\pi_P^\star) - V_P(\widehat{\pi}) &= \{V_P(\pi_P^\star) - V_P(\pi_n^\star)\} + \{V_P(\pi_n^\star) - V_P(\widehat{\pi})\} \\
&\leq \mathcal{A}_n(P; \Pi, \Pi_n) + \{V_P(\pi_n^\star) - \widehat{V}_{\mathrm{ls}}(\pi_n^\star)\} \\
&\quad + \{\widehat{V}_{\mathrm{ls}}(\pi_n^\star) - \widehat{V}_{\mathrm{ls}}(\widehat{\pi})\} + \{\widehat{V}_{\mathrm{ls}}(\widehat{\pi}) - V_P(\widehat{\pi})\} \\
&\leq \mathcal{A}_n(P; \Pi, \Pi_n) + 2 \sup_{\pi \in \Pi_n} \left| \widehat{V}_{\mathrm{ls}}(\pi) - V_P(\pi) \right| \\
&\quad + \frac{2\Delta_{\mathrm{cert}}}{n\varepsilon} \{\log M_n + \log(1/\beta)\} \\
&\leq \mathcal{A}_n(P; \Pi, \Pi_n) + 2C B_\psi \sqrt{\frac{\log(2M_n/\alpha)}{n}} + 2K\zeta^{-1} r_{\mu,m} r_{e,m} \\
&\quad + \frac{2\Delta_{\mathrm{cert}}}{n\varepsilon} \{\log M_n + \log(1/\beta)\}.
\end{aligned}$$

Renaming numerical constants as $C, C', C''$ yields (11). $\qquad\square$

*Proof of Corollary 1.* Apply Theorem 2. Under the stated assumptions, the library approximation term, the scoring-sample concentration term, and the doubly robust product term vanish. The privacy term also vanishes because

$$\frac{\Delta_{\mathrm{cert}}}{n\varepsilon} \{\log M_n + \log(1/\beta)\} = \max \left\{ \frac{2B_\psi}{n\varepsilon}, \frac{\rho_m^\star}{\varepsilon} \right\} \{\log M_n + \log(1/\beta)\} \to 0,$$

where the first component is controlled by $\log(M_n)/n \to 0$ and the second by the assumed stability condition. If $\rho_m^\star = O((m\lambda_m)^{-1})$, the displayed condition follows from $(\log M_n + \log(1/\beta))/(\varepsilon m \lambda_m) \to 0$. $\qquad\square$

# E Regularized ERM Stability Proof and Extensions

This appendix proves deterministic replace-one prediction stability for scalar regularized empirical risk minimization (ERM) and extends the result to multi-component and vector-valued nuisance learners. Proposition 2 gives the scalar bound. Corollary 2 records the extensions.

**Proposition 2.** *Let a scalar nuisance estimator solve the following regularized objective (with deterministic tie-breaking if necessary):*

$$\widehat{\theta}_S \in \arg \min_\theta \left\{ \frac{1}{m} \sum_{i \in S} \ell(\theta; Z_i) + \frac{\lambda_m}{2} \|\theta\|_2^2 \right\}.$$

*Assume the loss function $\ell(\theta; z)$ is convex and $G$-Lipschitz in $\theta$ for all $z$, which ensures the overall objective is $\lambda_m$-strongly convex. Further assume the scalar prediction map $g_\theta(x)$ is $L_g$-Lipschitz in $\theta$ uniformly over $x$. Then, for any adjacent training samples $S \sim S'$, the maximum pointwise deviation satisfies:*

$$\sup_x |g_{\widehat{\theta}_S}(x) - g_{\widehat{\theta}_{S'}}(x)| \leq \frac{2L_g G}{m \lambda_m}.$$

The proof of Proposition 2 is stated below. The Lipschitz loss assumption is used only through a uniform bound on the loss gradient along the parameter region in which the two adjacent minimizers lie. Hence, for a differentiable convex loss, the same proof applies whenever one can exhibit a public set $\Theta_m$ containing all possible minimizers and a public constant $G$ such that $\sup_{\theta \in \Theta_{m,z}} \|\nabla_\theta \ell(\theta; z)\|_2 \leq G$. This localized form is the one used for the ridge outcome learner in the ERM positive control below. Squared loss is not globally Lipschitz, but the bounded design and ridge penalty give a public norm bound on every fitted parameter vector, and therefore a public gradient bound on the relevant fitted domain.

**Corollary 2.** *Suppose the conditions of Proposition 2 hold.*

1. ***Coordinate-wise outcome learning and simplex propensity post-processing:*** *If each outcome component $\mu_a$ is learned by an independent scalar ERM learner whose objective is averaged over all $m$ records and then deterministically clipped to $[\underline{\mu}, \overline{\mu}]$, then*

$$\beta_{\mu,m} \leq \max_a \frac{2 L_{g,\mu,a} G_{\mu,a}}{m \lambda_{\mu,a}}.$$

    *If, in addition, $K$ coordinate-wise propensity score predictors $g_{e,a}$ are learned by independent scalar ERM objectives averaged over all $m$ records and the preliminary vector $g_e(x) = (g_{e,1}(x), \ldots, g_{e,K}(x))$ is deterministically post-processed as $\widehat{e}(x) = \mathcal{C}_\zeta(g_e(x)) \in \Delta_K^\zeta$, where $\mathcal{C}_\zeta$ is $L_{\mathcal{C},\infty}$-Lipschitz from $\ell_\infty$ to $\ell_\infty$, then*

$$\beta_{e,m} \leq L_{\mathcal{C},\infty} \max_a \frac{2 L_{g,e,a} G_{e,a}}{m \lambda_{e,a}}.$$

2. ***Vector-valued propensity learning:*** *Alternatively, suppose the multi-action propensity is learned jointly via a single vector ERM with a prediction map $g_\theta : \mathcal{X} \to \mathbb{R}^K$ satisfying $\sup_x \|g_\theta(x) - g_{\theta'}(x)\|_2 \leq L_{g,e} \|\theta - \theta'\|_2$. If the deterministic projection map $\mathcal{C}_\zeta : \mathbb{R}^K \to \Delta_K^\zeta$ used to produce $\widehat{e}(x) = \mathcal{C}_\zeta(g_{\widehat{\theta}}(x))$ is $L_{\mathcal{C}}$-Lipschitz from $\ell_2$ to $\ell_\infty$ uniformly in $x$, then the propensity prediction bound satisfies:*

$$\beta_{e,m} \leq \frac{2 L_{\mathcal{C}} L_{g,e} G_e}{m \lambda_e}.$$

The proof of Corollary 2 is stated below.

*Proof of Proposition 2 and Corollary 2.* We first prove the scalar stability statement of Proposition 2. Define the empirical objectives on the adjacent samples $S$ and $S'$ as:

$$F_S(\theta) = \frac{1}{m} \sum_{i \in S} \ell(\theta; Z_i) + \frac{\lambda_m}{2} \|\theta\|_2^2, \qquad F_{S'}(\theta) = \frac{1}{m} \sum_{i \in S'} \ell(\theta; Z_i) + \frac{\lambda_m}{2} \|\theta\|_2^2.$$

Let $\theta = \widehat{\theta}_S$ and $\theta' = \widehat{\theta}_{S'}$. By the $\lambda_m$-strong convexity of the objectives and the optimality of $\theta$ and $\theta'$, we obtain the inequalities:

$$F_S(\theta') \geq F_S(\theta) + \frac{\lambda_m}{2} \|\theta' - \theta\|_2^2, \qquad F_{S'}(\theta) \geq F_{S'}(\theta') + \frac{\lambda_m}{2} \|\theta' - \theta\|_2^2.$$

Adding these two inequalities yields:

$$\lambda_m \|\theta' - \theta\|_2^2 \leq \{F_S(\theta') - F_{S'}(\theta')\} + \{F_{S'}(\theta) - F_S(\theta)\}.$$

Because $S$ and $S'$ differ only by the replacement of a single record, say $z$ in $S$ replaced by $z'$ in $S'$, the regularization terms and the losses evaluated on all common records perfectly cancel. The right-hand side simplifies to:

$$\frac{1}{m} \{\ell(\theta'; z) - \ell(\theta'; z') + \ell(\theta; z') - \ell(\theta; z)\}.$$

Applying the $G$-Lipschitz continuity of the loss function $\ell$ with respect to $\theta$, we bound the differences:

$$\ell(\theta'; z) - \ell(\theta; z) \leq G \|\theta' - \theta\|_2, \qquad \ell(\theta; z') - \ell(\theta'; z') \leq G \|\theta' - \theta\|_2.$$

Substituting these bounds yields:

$$\lambda_m \|\theta' - \theta\|_2^2 \leq \frac{2G}{m} \|\theta' - \theta\|_2.$$

If $\theta' = \theta$, the desired bound holds trivially. Otherwise, dividing by $\|\theta' - \theta\|_2$ gives $\|\theta' - \theta\|_2 \leq 2G/(m\lambda_m)$. Finally, applying the $L_g$-Lipschitz continuity of the scalar prediction map $g_\theta(x)$ provides the scalar prediction bound:

$$\sup_x |g_{\widehat{\theta}_S}(x) - g_{\widehat{\theta}_{S'}}(x)| \leq L_g \|\widehat{\theta}_S - \widehat{\theta}_{S'}\|_2 \leq \frac{2L_g G}{m\lambda_m}.$$

We now prove the extensions detailed in Corollary 2. For coordinate-wise outcome learning, deterministic clipping to a fixed interval is a 1-Lipschitz operation. Applying the scalar stability result separately to each action-specific outcome learner immediately yields:

$$\max_a \sup_x |\widehat{\mu}_{S,a}(x) - \widehat{\mu}_{S',a}(x)| \leq \max_a \frac{2L_{g,\mu,a} G_{\mu,a}}{m\lambda_{\mu,a}}.$$

For coordinate-wise propensity learning, the scalar argument gives

$$\sup_x \|g_{e,S}(x) - g_{e,S'}(x)\|_\infty \leq \max_a \frac{2L_{g,e,a} G_{e,a}}{m\lambda_{e,a}}.$$

The released denominator vector is not obtained by independent coordinate clipping. It is the deterministic post-processing $\widehat{e}_S(x) = \mathcal{C}_\zeta(g_{e,S}(x)) \in \Delta_K^\zeta$. By the assumed $\ell_\infty$-to-$\ell_\infty$ Lipschitz property of $\mathcal{C}_\zeta$,

$$\max_a \sup_x |\widehat{e}_{S,a}(x) - \widehat{e}_{S',a}(x)| \leq L_{\mathcal{C},\infty} \max_a \frac{2L_{g,e,a} G_{e,a}}{m\lambda_{e,a}}.$$

The maximum over actions simply introduces the maximum of deterministic action-specific constants. No probabilistic union bound is required since the bound is deterministic and holds simultaneously for all $a \in \mathcal{A}$.

For the vector propensity implementation, the identical strong-convexity argument applies to the vector-valued objective, yielding $\|\widehat{\theta}_S - \widehat{\theta}_{S'}\|_2 \leq 2G_e/(m\lambda_e)$. The vector prediction Lipschitz condition then guarantees:

$$\sup_x \|g_{\widehat{\theta}_S}(x) - g_{\widehat{\theta}_{S'}}(x)\|_2 \leq \frac{2L_{g,e} G_e}{m\lambda_e}.$$

Because the deterministic post-processing map $\mathcal{C}_\zeta$ is $L_{\mathcal{C}}$-Lipschitz from $\ell_2$ to $\ell_\infty$ uniformly in $x$, we map this to the maximum element-wise deviation:

$$\max_a \sup_x |\widehat{e}_{S,a}(x) - \widehat{e}_{S',a}(x)| \leq \frac{2L_{\mathcal{C}} L_{g,e} G_e}{m\lambda_e}.$$

Thus, the scalar outcome learners and the multi-action vector propensity learner satisfy the deterministic bounds in (8). Substitution into (9) gives $\rho_m^\star = O((m\lambda_m)^{-1})$ when the primitive constants are bounded and the regularization levels have common order $\lambda_m$. $\qquad\square$

## F  Additional Example: Nearest-Neighbor Nuisance Sensitivity

Consider binary actions, a constant policy $\pi_1(x) \equiv 1$, fixed propensities $\widehat{e}_1(x) = \widehat{e}_2(x) = 1/2$, and outcomes in $[0, 1]$. Let $\widehat{\mu}_{S,1}$ be clipped one-nearest-neighbor regression using action-1 training records, and let $\widehat{\mu}_{S,2} = 0$. Construct adjacent samples $S \sim S'$ by replacing a single action-1 record at covariate $x_0$ with outcome 1 in $S$ and outcome 0 in $S'$, with all other action-1 covariates farther from $x_0$. Then

$$\sup_x |\widehat{\mu}_{S,1}(x) - \widehat{\mu}_{S',1}(x)| = 1.$$

Use a common scoring block with all $X_i = x_0$, $A_i = 2$, and $Y_i = 0$. Since $A_i \neq \pi_1(X_i)$, the inverse-propensity correction is zero and each DR score equals the learned action-1 mean. Therefore

$$|U_{\mathrm{ls}}(D, \pi_1) - U_{\mathrm{ls}}(D', \pi_1)| = n.$$

# G  Additional Experimental Details

## G.1  Experimental Setup Details

**ACIC preprocessing.**  We preprocess the ACIC 2016 covariates as follows. A raw column is treated as numeric if parseable numeric entries account for at least 90% of all rows. Otherwise it is treated as categorical. Missing numeric entries are imputed by the column median, and numeric columns are standardized to have zero mean and unit empirical standard deviation. Categorical columns are one-hot encoded, with missing values treated as an additional level. This produces 4,802 rows and 58 encoded covariates.

**Rare-region DGP.**  Let

$$h_R(x) = \sum_{j=1}^{\min\{8,d\}} c_j x_j + \mathbf{1}\{d > 15\}\{0.5\sin(x_8) + 0.25 x_{11} x_{15}\},$$

where $(c_1, \ldots, c_{\min\{8,d\}})$ is linearly spaced from 1.2 to $-0.9$. The rare-region indicator is

$$R(x) = \mathbf{1}\{h_R(x) \geq q_{0.95}\},$$

where $q_{0.95}$ is the empirical 95th percentile of the rare-region score on the ACIC covariate pool. For $k = \min\{10, d\}$, define sparse coefficient vectors

$$w_e = \text{linspace}(0.85, -0.65, k), \quad w_0 = \text{linspace}(0.55, -0.35, k), \quad w_\tau = \text{linspace}(-0.75, 0.95, k),$$

padded with zeros outside the first $k$ coordinates. The main rare-region separation audit uses

$$g_e(x) = 1.2 \frac{x^\top w_e}{\sqrt{k}} + 0.4\sin(x_1) - 0.25 x_2 \mathbf{1}\{x_3 > 0\},$$

$$g_0(x) = \frac{x^\top w_0}{\sqrt{k}} + 0.3\sin(x_1 x_4) + 0.2\mathbf{1}\{x_5 > 0\},$$

$$g_\tau(x) = \frac{x^\top w_\tau}{\sqrt{k}}.$$

The response surfaces used in the main text are those in Section 5.1. Standard samples use Gaussian noise with standard deviation 0.06. In the targeted adjacent-pair construction, non-anchor outcomes use standard deviation 0.04, and the rare anchor outcome is deterministically set to 1 in one fitting sample and 0 in its adjacent counterpart.

**Targeted adjacent-pair audit.**  The rare-region separation audit uses adjacent nuisance-fitting samples designed to expose the nuisance-fitting contribution to utility sensitivity. Each base fitting sample contains one rare-region anchor and $m - 1$ nonrare records. The adjacent sample keeps the same covariates and treatment assignments but flips the anchor outcome from 1 to 0. The stress scoring block is sampled from the rare pool. This construction changes one training record while evaluating the resulting learned score map on a scoring block concentrated in the low-mass region, matching the mechanism highlighted by Theorem 1.

For each audited adjacent pair, we compute

$$\widehat{\Delta}_{\text{move}} = \max_{\pi \in \Pi_n} |U_{\text{ls}}(D, \pi) - U_{\text{ls}}(D', \pi)|.$$

When a reference audit scale is needed in plots, we use the in-sample diagnostic

$$\widehat{\Delta}_{\text{audit}} = \max\{2B_\psi, 1.3\,\widehat{\Delta}_{\text{move}}\}.$$

Because this scale is constructed from the same observed adjacent-pair movement, its coverage of that pair is by construction and is not reported as an empirical result. Separately, on the same scoring block we compute

the empirical prediction movement

$$\widehat{\Gamma}_\mu = \max_{a \in \{0,1\}} \max_{x \in D^{sc}} |\widehat{\mu}_{S,a}(x) - \widehat{\mu}_{S',a}(x)|,$$

$$\widehat{\Gamma}_e = \max_{x \in D^{sc}} |\widehat{e}_S(x) - \widehat{e}_{S'}(x)|,$$

and the empirical decomposition proxy

$$\widehat{\rho}_{pred} = (1 + \zeta^{-1})\widehat{\Gamma}_\mu + \zeta^{-2}\widehat{\Gamma}_e, \qquad \widehat{\Delta}_{proxy} = \max\{2B_\psi, n\widehat{\rho}_{pred}\}.$$

Throughout the experimental appendix, "audit" denotes an empirical replace-one measurement, and formal calibration uses a public deterministic stability bound of the form (8).

**Policy library.** For the default library size $M = 160$, the public policy class consists of 18 structured policies and 142 random linear policies. The structured policies are:

$$\text{always treat}, \qquad \text{never treat},$$
$$\mathbf{1}\{x_j \geq 0\}, \quad \mathbf{1}\{x_j < 0\}, \qquad j = 1, \ldots, \min\{6, d\},$$
$$\mathbf{1}\{x^\top w_\tau \geq 0\}, \qquad \mathbf{1}\{x^\top w_\tau < 0\},$$
$$\mathbf{1}\{h_R(x) \geq q_{0.95}\}, \qquad \mathbf{1}\{h_R(x) < q_{0.95}\}.$$

The remaining policies are generated from public randomness. For each random linear rule, we sample a Gaussian direction. With probability 0.70, we retain a uniformly random subset of $\max\{2, \min\{8, d\}\}$ coordinates and zero out the rest. We then normalize the direction to unit Euclidean norm, draw an intercept from $N(0, 0.2^2)$, and use the threshold rule $\mathbf{1}\{x^\top w + b \geq 0\}$. The policy library is generated independently of the private sample.

**Nuisance learners.** Propensities are fit by logistic regression and clipped to $[\zeta, 1-\zeta]$. In the regularization sweeps, the logistic penalty is $\max\{0.1, \lambda\}$. Outcome regressions are fit separately for the two treatment arms using boosted-tree regressors with squared-error loss, subsampling and column subsampling set to one, one computational thread, and fixed public seeds. The regularized default learner uses depth 3, 100 trees, learning rate 0.06, and $\lambda = 6$. The unstable stress learner uses depth 7, 180 trees, learning rate 0.06, and $\lambda = 10^{-8}$. In the regularization audit, the adjacent pair, scoring block, and policy library are held fixed across values of $\lambda$ so that changes in movement are attributable to learner regularization.

**ERM positive control.** The positive control uses public preprocessing and deterministic strongly convex ERM nuisance learners with public stability constants. Every covariate is deterministically clipped to $[-3, 3]$ and rescaled to $[-1, 1]$ before fitting. This preprocessing gives the feature vector with intercept the deterministic radius $R_x = \sqrt{d+1}$, with $d = 20$ in the reported run. For each treatment arm, the outcome nuisance solves the ridge objective for the centered residual $Y - 1/2$,

$$\frac{1}{m} \sum_{i=1}^m \mathbf{1}\{A_i = a\} \frac{1}{2} \{\theta_a^\top \widetilde{X}_i - (Y_i - 1/2)\}^2 + \frac{\lambda_\mu}{2} \|\theta_a\|_2^2,$$

and predicts $\text{clip}_{[0,1]}\{1/2 + \theta_a^\top \widetilde{X}\}$. The propensity nuisance solves the $\ell_2$-regularized logistic ERM with fixed Newton/backtracking optimization and prediction clipping to $[\zeta, 1-\zeta]$. We use $\lambda_\mu = \lambda_e = 50$, $m = n = 1500$, $\zeta = 0.10$, and the same public policy library construction as in the main experiments.

With the bounded design, the public prediction-stability constants used in this experiment are

$$\beta_{\mu,m} = \frac{2L_{g,\mu}G_\mu}{m\lambda_\mu}, \qquad \beta_{e,m} = \frac{2L_{g,e}G_e}{m\lambda_e}.$$

Here the outcome constant is not obtained from a global Lipschitz property of squared loss. Let $r = Y - 1/2$, so $|r| \leq 1/2$. For any possible outcome training sample, comparing the ridge objective at its minimizer $\hat{\theta}_a$

with the objective at 0 gives

$$\frac{\lambda_\mu}{2}\|\hat\theta_a\|_2^2 \le \frac{1}{m}\sum_{i=1}^m \mathbf{1}\{A_i = a\}\frac{1}{2}r_i^2 \le \frac{1}{8}, \qquad \text{so} \qquad \|\hat\theta_a\|_2 \le \frac{1}{2\sqrt{\lambda_\mu}}.$$

For the per-record squared residual loss, $\nabla_\theta \ell_a(\theta; Z) = \mathbf{1}\{A = a\}(\theta^\top \widetilde X - r)\widetilde X$. Since $\|\widetilde X\|_2 \le R_x$, every fitted minimizer satisfies the uniform gradient bound

$$\|\nabla_\theta \ell_a(\hat\theta_a; Z)\|_2 \le R_x\left(\frac{1}{2} + \frac{R_x}{2\sqrt{\lambda_\mu}}\right) =: G_\mu.$$

The prediction map $\theta \mapsto 1/2 + \theta^\top \widetilde X$, followed by clipping to $[0,1]$, is $L_{g,\mu} = R_x$-Lipschitz in $\theta$. The localized-gradient version of Proposition 2 therefore yields the displayed $\beta_{\mu,m}$. For the logistic propensity learner, the regularized logistic loss has gradient norm at most $R_x$, and the clipped sigmoid prediction map is $R_x/4$-Lipschitz in $\theta$, and hence $G_e = R_x$ and $L_{g,e} = R_x/4$. The resulting deterministic prediction-movement radius is

$$\rho_m^\star = (1 + \zeta^{-1})\beta_{\mu,m} + \zeta^{-2}\beta_{e,m}, \qquad \Delta_{\text{cert}} = \max\{2B_\psi, n\rho_m^\star\}.$$

The empirical replace-one movement and likelihood-ratio audits reported below are diagnostics only. The calibration scale itself is the public deterministic quantity $\Delta_{\text{cert}}$.

**Experiment grid and reported quantities.** The main repetitions use $m = 1500$ fitting records, $n = 1500$ scoring records, library size $M = 160$, and 100 repetitions. The separation audit varies $n \in \{300, 700, 1200, 2000, 3000\}$. The regularization audit uses

$$\lambda \in \{0.01, 0.03, 0.1, 0.3, 1, 3, 10, 30, 100\}.$$

The sample-split decomposition fixes total sample size 3600 and varies the training fraction over

$$\{0.20,\ 0.25,\ 0.35,\ 0.50,\ 0.65,\ 0.75,\ 0.80\}.$$

Additional robustness outputs generated by the code vary the DGP regime, overlap level $\zeta$, dimension $d$, policy-library size, and privacy budget. The ERM positive-control experiment uses 100 repetitions with $m = n = 1500$, $d = 20$, $\lambda_\mu = \lambda_e = 50$, and reports the public stability quantities together with empirical adjacent-pair diagnostics.

### G.2  Column Definitions for Tables 1–3 and 5

All three tables report averages over 100 repetitions. In Table 1, "nuis. $L_2$" is nuisance RMSE; "DR prod." is the empirical doubly robust product-remainder proxy; "floor" is the fixed-score sensitivity $2B_\psi$; "unstable move/floor" is $\widehat\Delta_{\text{move}}/(2B_\psi)$ for the unstable learner; "naive logLR/$\varepsilon$" is the audited maximum log-likelihood ratio for the learned-score exponential mechanism calibrated with the fixed-score floor, divided by $\varepsilon$; and "audit-cal. logLR/$\varepsilon$" is the same in-sample audit under the scale constructed from the observed movement. The latter is a sanity check rather than independent validation.

In Table 2, "$\widehat\rho_{\text{pred}}$" is the empirical replace-one prediction-movement proxy; "move" is $\widehat\Delta_{\text{move}} = \max_{\pi \in \Pi_n} |U_{\text{ls}}(D, \pi) - U_{\text{ls}}(D', \pi)|$; "audit $\Delta$" is the unfloored audit scale $1.3\widehat\Delta_{\text{move}}$; "audit/floor" is that scale divided by $2B_\psi$; "floored $\Delta$" is $\max\{2B_\psi, 1.3\widehat\Delta_{\text{move}}\}$ in each repetition; "floor rate" is the fraction of repetitions in which $2B_\psi$ is active; and "regret" is expected exponential-mechanism regret.

In Table 3, "fit frac." is $m/(m + n)$; "nuis. RMSE" is nuisance RMSE; "$\widehat\rho_{\text{pred}}$", "move", "floored $\Delta$", "floor rate", and "regret" have the same meanings as in Table 2; and "$1/\sqrt{n}$" is the scoring-sample concentration scale from the regret decomposition.

In Table 5, $\rho_m^\star$ is the deterministic prediction-movement radius computed from the bounded-design ERM constants, "emp. $\rho$" is the sampled prediction-movement proxy, "$\Delta_{\text{cert}}$" is the deterministic sensitivity scale,

Table 4: Theorem-proxy and sampled audit scales over 100 repetitions.

| setting | move | audit $\Delta$ | floored $\Delta$ | proxy $\widehat{\Delta}$ | proxy/floored | floor rate |
|---------|------|----------|-----------|---------|---------------|------------|
| 0.01 | 40.18 | 52.23 | 54.70 | 13300 | 367.7 | 0.23 |
| 0.03 | 39.72 | 51.63 | 54.13 | 13300 | 368.3 | 0.26 |
| 0.10 | 39.27 | 51.05 | 53.55 | 13100 | 366.3 | 0.23 |
| 0.30 | 37.39 | 48.60 | 51.11 | 12700 | 364.2 | 0.24 |
| 1.00 | 33.94 | 44.12 | 46.55 | 11400 | 343.5 | 0.25 |
| 3.00 | 25.41 | 33.04 | 36.15 | 6900 | 237.0 | 0.41 |
| 10.0 | 17.54 | 22.80 | 27.76 | 3600 | 140.1 | 0.57 |
| 30.0 | 12.94 | 16.82 | 23.90 | 2100 | 89.45 | 0.77 |
| 100 | 5.15 | 6.70 | 22.00 | 719.0 | 32.68 | 1.00 |
| 0.20 | 2.61 | 3.39 | 22.00 | 26000 | 1200 | 1.00 |
| 0.25 | 2.12 | 2.76 | 22.00 | 17900 | 811.5 | 1.00 |
| 0.35 | 1.41 | 1.83 | 22.00 | 10800 | 490.9 | 1.00 |
| 0.50 | 1.01 | 1.31 | 22.00 | 5300 | 239.3 | 1.00 |
| 0.65 | 0.73 | 0.95 | 22.00 | 3400 | 154.2 | 1.00 |
| 0.75 | 0.56 | 0.73 | 22.00 | 1700 | 77.24 | 1.00 |
| 0.80 | 0.51 | 0.66 | 22.00 | 1300 | 58.91 | 1.00 |

"move/$\Delta_{\mathrm{cert}}$" is the observed utility movement divided by $\Delta_{\mathrm{cert}}$, "logLR/$\varepsilon$" is the empirical privacy-loss audit divided by the target privacy level, "cover" is the fraction of audited adjacent pairs covered by $\Delta_{\mathrm{cert}}$, "regret" is expected exponential-mechanism regret under $\Delta_{\mathrm{cert}}$, and "nuis. RMSE" is the average nuisance RMSE across the two outcome nuisances and propensity nuisance.

For statistical diagnostics, we report nuisance RMSE for $\widehat{\mu}_0, \widehat{\mu}_1, \widehat{e}$, the DR product proxy

$$\widehat{r}_{\mathrm{DR}} = \frac{1}{2}\{\mathrm{RMSE}(\widehat{\mu}_0) + \mathrm{RMSE}(\widehat{\mu}_1)\}\mathrm{RMSE}(\widehat{e}),$$

and empirical regret. For exponential-mechanism policy-selection mechanisms, regret is the expectation under the mechanism's selection distribution rather than a single draw from the mechanism. For sensitivity diagnostics, we report the observed utility movement $\widehat{\Delta}_{\mathrm{move}}$, the audit scale $\widehat{\Delta}_{\mathrm{audit}}$, the prediction-movement proxy $\widehat{\rho}_{\mathrm{pred}}$, and, in the output files, the decomposition scale $\widehat{\Delta}_{\mathrm{proxy}}$.

### G.3 Diagnostic Gap Between Sampled Audits and Theorem-Style Proxies

Table 4 compares the sampled adjacent-pair audit scale with the algebraic theorem-proxy scale

$$\widehat{\Delta}_{\mathrm{proxy}} = \max\{2B_\psi, n\widehat{\rho}_{\mathrm{pred}}\}.$$

For boosted-tree nuisances, $\widehat{\rho}_{\mathrm{pred}}$ is an empirical prediction-movement proxy measured on sampled adjacent pairs. Therefore $\widehat{\Delta}_{\mathrm{proxy}}$ is a theorem-shaped diagnostic scale obtained by substituting observed movement into the sensitivity decomposition; formal pure-DP calibration requires a public deterministic upper bound of the form (8).

The comparison highlights two points that are useful for interpreting the main experimental claims. First, the empirical audit movement and audit scale decrease with stronger regularization and with larger nuisance-fitting fractions, matching the qualitative direction of the sensitivity decomposition. Second, the plug-in theorem-proxy scale remains substantially larger than the sampled audit scale for boosted trees. This gap is

Table 5: ERM positive-control diagnostic over 100 repetitions. Appendix G.2 states column definitions.

| $\rho_m^\star$ | emp. $\rho$ | $\rho_m^\star$/emp. $\rho$ | $\Delta_{\mathrm{cert}}$ | move/$\Delta_{\mathrm{cert}}$ | logLR/$\varepsilon$ | cover | regret | nuis. RMSE |
|---|---|---|---|---|---|---|---|---|
| 0.0191 | $7.49{\times}10^{-4}$ | 26.87 | 28.61 | $1.25{\times}10^{-5}$ | $4.88{\times}10^{-6}$ | 1.00 | 0.0146 | 0.0423 |

expected: the audit scale measures observed movement over sampled adjacent pairs, whereas theorem-level pure DP requires a public deterministic upper bound that holds uniformly over adjacent datasets. Thus the boosted-tree experiments provide stress-test diagnostics for the nuisance-fitting sensitivity mechanism; Theorem 2 applies when the empirical proxy is replaced by a public deterministic stability bound.

### G.4  ERM Positive Control

Table 5 reports the ERM positive control described in Appendix G.1. The experiment uses ACIC covariates with public clipping and rescaling, $d = 20$, $m = n = 1500$, $\lambda_\mu = \lambda_e = 50$, $\zeta = 0.10$, and 100 repetitions. The calibration uses the public deterministic scale $\Delta_{\mathrm{cert}} = \max\{2B_\psi, n\rho_m^\star\}$ computed from the bounded-design ERM constants.

The ERM positive control distinguishes validity from tightness. The deterministic prediction-movement radius $\rho_m^\star = 0.0191$ is about 27 times the sampled prediction-movement proxy, yielding $\Delta_{\mathrm{cert}} = 28.61$. The bound is valid for the stated learner, but the gap shows that it is conservative. All audited adjacent movements are covered by this scale. The mean observed movement divided by $\Delta_{\mathrm{cert}}$ is $1.25 \times 10^{-5}$, and the maximum over repetitions is $2.36 \times 10^{-5}$. The empirical likelihood-ratio audit is also far below the privacy target, with mean $\mathrm{logLR}/\varepsilon = 4.88 \times 10^{-6}$ and maximum $9.57 \times 10^{-6}$. At the same time, selection calibrated with this deterministic scale retains useful statistical behavior: the average nuisance RMSE is 0.0423 and the expected regret under $\Delta_{\mathrm{cert}}$ is 0.0146.

In summary, the boosted-tree experiments diagnose the nuisance-fitting sensitivity mechanism, while the ERM positive control instantiates the public stability bound with deterministic strongly convex learners on the same ACIC covariate design.

