# OpenReview forum: "Global Sensitivity in Private Policy Selection with Nuisance-Fitted Doubly Robust Scores"
_TMLR — Under review for TMLR_

### Review · Reviewer_X92m · 2026-06-05

**Summary Of Contributions:**

This paper demonstrates that doubly robust (DR) estimators, a widely used estimator in causal inference for their statistical resilience to nuisance model misspecification, do not provide the worst-case individual-level stability required for valid differential privacy (DP). The authors formally prove that naively plugging learned DR policy-value scores into the exponential mechanism for private policy selection can catastrophically violate $\\epsilon$-DP even when all standard DR diagnostics (nuisance $L_2$ error, product remainder, population bias) indicate excellent performance. The paper then provide the error rate for certified selection mechanism under DP guarantee. Numerical results verified the theoretical analysis on sensitivity and error rate.

Strength:
1. This paper provides novel theoretical analyses on how robustness in causal inference and policy evaluation differ from DP sensitivity and privacy measurement. DR studies average robustness while DP focuses on worst-case robustness. The paper also provides the theoretical analysis on the regret of the algorithm and how it decompose into different factors.

2. The paper provides numerical evaluations to validate the theory.

3. The paper is well-structured. The notations and assumptions are clearly defined and stated. Numerical results are also clearly explained.

Weakness:
1.  Theroem 2 uses pure $\\epsilon$-DP, but still involves "with probability $1-\\delta$" in the statement. This could be reformat as approximate DP, i.e., $(\\epsilon, \\delta)$-DP.
2. The problem is restricted to public data-independent policy library. It cannot be applied to continuous policy sets, such as policies models by neural networks, and mordern policy optimization models.
3. The bound provid by the paper is quite loose in the numerical results. As stated in G.4, the certified $\\rho$ is ~27 times the empirical sensitivity. This bound is impractical for real-world applications.

**Audience:**

Yes

**Audience Explanation:**

1. The paper provides novel analysis on the robustness in DR and DP. It provides an initial understanding on the connection and difference between these two measurements.

2. The paper provides the decomposition of the regret under a set of restrictive assumptions. This is also an initial step to better understanding and designing new policy optimiation algorithms under DP and DR restrictions.

However, the scope of this paper is still limited and requres future research.

**Claims And Evidence:**

Yes

**Claims Explanation:**

1. The paper is clearly written.
2. The theoretical analysis and the proofs are provided in the main paper and in the appendix. There is no obvious error in the proof steps.
3. Numerical results matches the theoretical analysis, showing the orthorgonality between DR robustness and privacy. Experiment setting are included in the appendix.

**Requested Changes:**

1. Please address the privacy measurement in theorem 2 ($\\epsilon$-DP to $(\\epsilon, \\delta)$-DP).

2. Add certain comment on the restriciton on problem class. It would be helpful if the authors can provide some insight on how the results could be potentially expand to continuous policies.

---

### Review · Reviewer_NBWf · 2026-06-12

**Summary Of Contributions:**

The paper studies the interplay of differential privacy (DP) and doubly robust estimation (DR) in the context of private policy selection. They consider what would happen if one tried to combine, in a natural way, DP (specifically the exponential mechanism) and doubly-robust estimation techniques for the purposes of selecting the best-utility policy from a finite (public) candidate set of policies --- and exhibit difficulties with that idea, showing that the pipeline is not seamless. Specifically, they introduce the notion of "sensitivity spillover", which consists in the fact that perturbing one record can affect the validity of evaluating scores on other records. On the positive side, then, they give a certificate, "public deterministic replace-one prediction stability", that when satisfied ensures the validity of the DP/DR pipeline as a whole, and give a regret-style theorem with several distinct components in the bound that formalizes that. Finally, the conducted experiments, which are semi-synthetic, confirm the effects that the developed theory predicts: that in private policy estimation, one can observe substantial "sensitivity spillover", and that this can be ameliorated with stability-oriented regularization.

**Additional Comments:**

################# UPDATE: Having familiarized myself with the authors response to my review, and to the extensive and well-executed edits made in the submission revision, I have now changed the Audience criterion response to "Yes". At this point, the revised submission positions its motivating questions and resulting contributions much more clearly than before, and will now be more readable/parseable to its intended audience at the cusp of differential privacy and causality/doubly robust estimation/policy selection.

The updated exposition substantially clarifies the merit of studying the pitfalls and guarantees of combining doubly-robust estimation with exponential-mechanism DP in pipelines where the best policy (out of a finite set) has to be selected with privacy guarantees.

In this way, my above complaints have been substantively addressed, both in terms of the motivation of the work and its clarity of exposition.
(Additionally, the other reviewers' clarifying remarks and questions were also, in my opinion, taken into account by the authors in a way that further improves clarify, positioning relative to prior work, and the ramifications of the conservativeness of the bounds.)

**Audience:**

Yes

**Audience Explanation:**

On this criterion, I am not satisfied with the current state of the paper. This is for two main reasons, which overlap with the preceding discussion.

1. To garner the interest of some subset of TMLR readers, one would hope that the paper would take up questions that are both useful, and not a-priori clear how to answer. Additionally, one would hope that the questions under study, as well as the answers provided by the authors, would offer some amount of “surprisal” to the intended audience. In this particular case, the paper studies the relationship between DP and DR estimation for the purposes of selecting the best policy, and — at a high level — offers the takeaway that the two do not easily mix (without further assumptions/certificates/etc.). To quote the very first sentence of the manuscript itself, however, “Doubly robust (DR) scores are a standard method in causal inference and policy evaluation: they make policy-value estimates insensitive to global nuisance estimation error. Differential privacy (DP), however, requires a different form of robustness: the released output must be stable under the replacement of any one individual.” In some sense, one can already pause here and ask: sure, of course — both notions are, under some natural interpretation, about *robustness* — but they very much differ in the exact meaning and context of the word robustness. And when the paper underscores that the connection between DP and DR in the context of policy selection is *not seamless*, the same question arises: why would it be? More generally, when one wishes to study concepts/notions X and Y and compare and contrast them against each other, or examine their interrelationship, it would be nice to understand the *synergies* between the two concepts/notions. One of the takeaways of this paper, however, seems to be that controlling the effects/guarantees of DP and DR is a roughly speaking orthogonal pair of affairs.
2. The second point is that the above argument casting doubt on the inherent degree of interest that the paper might evoke in readers can be objected to: perhaps the related literature has already posed questions about the synergies between DP and DR? However, other than providing a short laundry list of papers in the intersection of causal inference and differential privacy, the paper does little to position itself as meaningfully extending those papers’ narratives or complementing them in productive ways. The only type of positioning that I saw was of the form “we are the first to study the particular question of private policy selection under double robustness and …” This does argue for the novelty of this paper, but not for the inherent interest that that might/should invoke. (Indeed, one cannot use the weaker standard of “differential privacy is important, and doubly robust estimation is important, and policy selection is an important problem, hence a paper that studies them together must be important”).

**Claims And Evidence:**

Yes

**Claims Explanation:**

On the correctness and evidence criterion, my evaluation is mixed. First of all, to the extent that I’ve carefully read the main part of the paper and the appendix, I believe that: (1) the mathematical side of the paper is substantially correct (indeed, the theory side of this paper is fairly straightforward, using basic inequalities and concentration bounds and never steering too far from the definitional properties of the basic objects of study — DP (the exponential mechanism) and DR estimators), and the correctness check is admittedly a rather basic requirement to check off in this paper’s instance); and (2) the experiments conducted by the authors are sufficiently well-executed and provide a useful sanity check for the theory. As such, the manuscript can be said to satisfy the correctness aspect of this criterion.

However, in terms of the facet of clarity, the paper cannot in my opinion lay a serious claim to being well-written. Fortunately, this primarily concerns its prose rather than the mathematical expressions/displays, but unfortunately this made it a very, and unnecessarily, hard read for me — and I believe for many prospective readers too. Indeed, the writing exhibits some of the worse traits of ChatGPT-style prose (note that I do not wish to make any claims as to whether or not this is how this manuscript was actually written/edited). Instead of clear and concise explanations, weird terminology and lengthy, winding and laundry-list-like sentences are ubiquitous.

For instance, the following chunk appears in the abstract (!) which one would have hoped would clearly, and briefly, convey the nature of the contributions of the paper, but instead we get: “We prove a separation showing that double robustness, vanishing nuisance error, and even zero DR population bias can coexist with order-n realized utility sensitivity, invalidating the usual fixed-utility privacy calibration. We then give a sufficient certification route based on deterministic replace-one prediction stability of the nuisance learners, which yields a valid pure-DP exponential mechanism and a regret bound separating library approximation, concentration, DR product remainder, and certified privacy cost.” First of all, we just see laundry lists of items here. Second, I doubt many readers would understand what the following terms mean, without reading a substantial chunk of the paper: “order-n realized utility sensitivity”, “library approximation”, “the usual fixed-utility privacy calibration”, “sufficient certification route”.

Similarly, many of the key terms used in the paper are confusing or non-standard. E.g. for a central example of this, “sensitivity spillover” is arguably not a good term to use from a causal inference literature perspective: spillover has a particular meaning related to the effects of treatment, and analogizing based on the fact that it talks about the impact on unrelated units is confusing at the very least.

To sum up, without a very significant and all-encompassing overhaul of the paper’s writing/prose, it is hard to imagine that most readers, from both the DP and the causal inference side, would find the paper clear and convincing.

**Requested Changes:**

The above discussion should be sufficient for a detailed understanding of my below requests (all of which are critical):

1. The paper must be properly positioned and motivated relative to the related work, and said related work (on DP + DR, and related settings) must be insightfully discussed rather than listed up.
2. The paper's contributions currently appear rote, and any potentially surprising and nontrivial takeaways must be properly highlighted. Again, currently DP and DR considerations largely interoperate --- depending on one's perspective --- in ways that are somewhat detached from each other.
3. The paper's writing/prose is not acceptable at this point, from my perspective; see above for examples and clarifications on what I mean by that. To convince me that the whole paper can and will be substantially improved/revised before it can be accepted, I would like to see a revamped and clear abstract and introduction, with much tighter prose and mechanics, as well as a clear plan for how the rest of the paper will be revised in that vein.

---

### Review · Reviewer_ygzX · 2026-06-15

**Summary Of Contributions:**

The paper studies private policy selection from a finite public policy library using learned doubly robust (DR) utilities. Its main point is that DR robustness and differential privacy robustness are different objects: DR controls population-level nuisance error, while pure DP requires worst-case stability of the realized utility under one-record replacement. The paper identifies a "sensitivity spillover" effect, where changing one record in the nuisance-training block can change the fitted score map and thereby affect all records in the scoring block. It proves a separation showing that vanishing nuisance error and zero DR population bias can coexist with order-(n) utility sensitivity. It then gives a sufficient certification route based on deterministic replace-one prediction stability of the nuisance learners, yielding a valid exponential-mechanism calibration and a regret decomposition.

**Strengths:** the central distinction is important and clearly motivated; the fixed/frozen/learned utility regimes are useful; the separation theorem is clean; and the regret decomposition separates statistical and privacy costs in a helpful way.

**Weaknesses:** the positive result relies on strong deterministic stability certificates that are likely not currently available for many flexible learners used in practice; some novelty claims should be stated more carefully relative to standard DP sensitivity/stability ideas; and the experiments are mainly diagnostic audits rather than privacy certificates, except for the ERM positive control.

**Audience:**

Yes

**Audience Explanation:**

Yes. The paper addresses what could be a real pitfall at the intersection of causal inference, policy learning, and differential privacy. In theory, readers working with DR scores may be tempted to treat bounded DR utilities as immediately compatible with the exponential mechanism. This paper gives a clear reason why that can be wrong when nuisances are learned on protected data.

The audience is somewhat specialized, since the paper focuses on finite public-library policy selection rather than general private policy learning. Still, the warning and certification principle should be of interest to researchers in private causal inference and offline policy learning.

**Broader Impact Concerns:**

N/A.

**Claims And Evidence:**

Yes

**Claims Explanation:**

Mostly yes. The theoretical claims are well aligned with the setup: for fixed or frozen utilities, bounded summands give the usual constant sensitivity, while for internally learned DR utilities the nuisance learner can induce score-map movement that accumulates over the scoring block. The separation result convincingly shows that standard DR guarantees do not imply pure-DP sensitivity control. The certification theorem is also sound as a sufficient condition, provided the deterministic replace-one stability certificate is available.

The empirical evidence supports the qualitative story: the reported audits show cases where nuisance/DR diagnostics remain benign while adjacent-dataset utility grows, and regularization reduces empirical movement. However, the paper should be more explicit that most experiments with flexible learners are not formal DP guarantees. They are useful stress tests, but a sampled adjacent-pair audit is not a substitute for a global sensitivity bound. The ERM positive-control experiment is the part that most directly supports the certification analysis.

**Requested Changes:**

1. Temper the novelty claim/language. The application to learned DR utilities for private policy selection appears novel and useful, but the general observation that data-dependent learned objectives can have larger global sensitivity is familiar in DP, e.g., SGD. The novelty is the causal policy selection instantiation, the spillover decomposition, and the DR-specific separation.

2. Separate audits from certificates more clearly. The experimental sections should repeatedly distinguish between empirical adjacent-pair movement audits and formal DP sensitivity certificates. In particular, the XGBoost-style experiments should not be described in language that could be read as certifying privacy.

3. More optional: Discuss alternative privacy routes in more details. A brief comparison to privately trained nuisances, private ERM throughout, smooth/local sensitivity under approximate DP, or privatizing only released value estimates would better position the proposed certificate route.